# Parameterizing spectral surface reflectance relationships for the Dark Target aerosol algorithm applied to a geostationary imager

Mijin Kim[1,2], Robert C. Levy[2], Lorraine A. Remer[3], Shana Mattoo[2,4] and Pawan Gupta[2]

[1]Goddard Earth Sciences Technology and Research (GESTAR) II, Morgan state university, Baltimore, MD 21251, USA
5   [2]NASA Goddard Space Flight Center, Greenbelt, MD 20771, USA
[3]Goddard Earth Sciences Technology and Research (GESTAR) II, University of Maryland, Baltimore County, Baltimore, MD 21250, USA
[4]Science Systems and Applications (SSAI), Lanham, MD 20706, USA

*Correspondence to*: Mijin Kim (mijin.kim@nasa.gov)

10   **Abstract**. Originally developed for the Moderate Resolution Imaging Spectroradiometer (MODIS) in polar, sun-synchronous low-earth orbit (LEO), the Dark Target (DT) aerosol retrieval algorithm relies on the assumption of a Surface Reflectance Parameterization (SRP) over land surfaces. Specifically for vegetated and dark-soiled surfaces, values of surface reflectance in blue and red visible-wavelength bands are assumed to be nearly linearly related to each other and to the value in a shortwave infrared (SWIR) wavelength band. This SRP also includes dependencies on scattering angle and a normalized difference vegetation index computed from two SWIR bands (NDVI$_{SWIR}$). As the DT retrieval algorithm is being ported to new sensors to continue and expand the aerosol data record, we assess whether the MODIS-assumed SRP can be used for these sensors. Here, we specifically assess SRP for the Advanced Baseline Imager (ABI) aboard, the Geostationary Operational Environmental Satellite (GOES)-16/East (ABIE). First, we find that using MODIS-based SRP leads to higher biases and artificial diurnal signatures in aerosol optical depth (AOD) retrievals from ABIE. The primary reason appears to be that geostationary orbit (GEO) encounters an entirely different set of observation geometry than does LEO, primarily with regards to solar angles coupled with fixed view angles. Therefore, we have developed a new SRP for GEO that draws the angular shape of the surface bidirectional reflectance. We also introduce modifications to the parametrization of both red-SWIR and blue-red spectral relationships to include additional information. The revised Red-SWIR SRP includes solar zenith angle, NDVI$_{SWIR}$, and land-type percentage from an ancillary database. The blue-red SRP adds dependencies on the scattering angle and NDVI$_{SWIR}$. The new SRPs improve the AOD retrieval of ABIE in terms of overall less bias and mitigation of the overestimation around local noon. The average bias of DT AOD compared to AERONET AOD shows a reduction from 0.08 to 0.03, while the bias of local solar noon decreases from 0.12 to 0.03. The agreement between DT and AERONET AOD is established through regression slope of 1.06 and y-intercept of 0.01 with correlation coefficient is 0.74. By using the new SRP, the percentage of data falling within the expected error range (±0.05 + 15%) is notably risen from 54% to 78%.

## 1. Introduction

Aerosols in the atmosphere strongly influence the Earth's energy budget by absorbing and scattering solar radiation, and by acting as cloud condensation nuclei and ice nuclei to alter cloud micro- and macrophysics (Boucher et al., 2014). They also play an important role in global atmospheric chemistry and the biogeochemical cycle (Kanakidou et al., 2018), and affect local air quality as particulate matter. Satellite remote sensing is beneficial to monitor atmospheric aerosol on the global scale. Broad coverage allows capturing widespread distribution and long-range transport of aerosol (Yu et al., 2013), and can help determine aerosol effects on climate and air quality.

Taking advantage of satellite observations, numerous efforts have been made to infer global distributions of aerosol optical properties. The procedure to infer aerosol loading and characteristics over ocean takes advantage of the relatively well-known surface reflectance properties of the ocean surface. It is much more difficult to do the same over land because the surface is both variable and dynamic. Over 25 years ago, Kaufman and Remer (1994) and Kaufman et al. (1997b) noted that for vegetation and dark-soiled surfaces, the land surface reflectance in some visible (VIS) wavelengths were highly correlated with each other, and also with values measured in some shortwave infrared (SWIR) wavelengths. This led to an aerosol retrieval approach over land used for MODIS (e.g., Kaufman et al., 1997a), for which after modifications is known as the 'Dark Target (DT) aerosol algorithm' (Remer et al., 2008, 2005; Levy et al., 2013, 2007a, b, 2010, 2013).

In the original version (Kaufman et al., 1997a), the surface reflectance parameterization (SRP) assumed that the values in the 0.47 μm blue and the 0.65 μm red bands were set to be ¼ and ½ the values in the 2.11 μm SWIR band. With the current MODIS version, instead of simple ratios, the spectral relationship includes the slopes and intercepts of their regressions. In addition, the current SRP includes dependencies on scene identification. It was noted during the first years of MODIS on-orbit data collection that bidirectional reflectance introduced angular dependencies in the SRPs, and these dependencies were parameterized in subsequent algorithm versions as a function of scattering angle (Levy et al., 2007b; Remer et al., 2001). This DT retrieval algorithm has led to derivation of a 20+ yearlong AOD record that has an outstanding performance and relied on by an extensive user community. Highly accurate aerosol products of the DT algorithm have not only contributed to improving theoretical understanding about the role of aerosols in radiation and climate (e.g., Boucher et al., 2014), but they have also been used for monitoring surface air quality (e.g., Al-Saadi et al., 2005; Chu et al., 2003) and a variety of other applications (Remer et al., 2020).

In order to continue the aerosol data record after the decommissioning of the MODIS missions on both Terra and Aqua, the DT algorithm is being ported to other sensors. For example, Sawyer et al., (2020) discuss porting DT to the Visible-infrared Radiometer Suite (VIIRS) which is also on polar-orbiting satellites in Low Earth Orbit (LEO). The new advanced imagers observe a similar spectral range (visible through thermal infrared) as MODIS, but there are generally fewer bands in total, and the wavelength range of analogous bands are shifted. Levy et al. (2015) and Sawyer et al. (2020) showed that with minimal change to SRP and slight adjustments for wavelength band, one could derive a VIIRS AOD product that had error/bias statistics

similar to the MODIS product. Presumably, only small changes were necessary because VIIRS offers similar observing geometry to MODIS (i.e., both MODIS and VIIRS pass over the equator ±1.5 hours around noon).

While retrievals from LEO provide near-daily global aerosol products, their low frequency of observations limits the understanding of rapid aerosol changes. In contrast, continuous imaging by sensors in geostationary orbit (GEO) can capture diurnal variations of aerosol occurring with human activity cycles, outbreak of emission episodes, and long-range transport (Kim et al., 2020). Saide et al., 2014 showed that diurnal variations of AOD captured by the Geostationary Ocean Color Imager (GOCI) on board the Geostationary Korea Multi-Purpose Satellite (GK)-1 (Choi et al., 2018) significantly improve surface air quality simulation in Korea in comparison to only assimilating MODIS DT AOD.

Many imagers in GEO now include the capability to measure both visible and SWIR reflectance, the DT algorithm can be adapted to advanced imagers in GEO, such as the Advanced Himawari Imager (AHI) on board the Japanese Himawari satellite series (currently Himawari-8 and -9) and the Advanced Baseline Imager (ABI) on board National Oceanic and Atmospheric Administration's (NOAA) operational Geostationary Operational Environmental Satellite (GOES) series (GOES-16, -17 and -18). Already, NOAA is using a DT-like approach to retrieve aerosol products from ABI (Laszlo et al. 2022, Zhang et al. 2020), and a DT-SRP is being used within the Yonsei Aerosol Retrieval (YAER) algorithm to derive AOD from AHI (Lim et al., 2018).

The NASA DT algorithm has in fact been ported to a geostationary orbit (GEO) sensor (Gupta et al., 2019) in order to test a GEO-LEO synergy concept [Making Earth System Data Records for Use in Research Environments; MEaSUREs Program, https://www.earthdata.nasa.gov/esds/competitive-programs/measures/leo-geo-synergy]. This initial (baseline) version of ABI's DT-algorithm accounts for shifted wavelengths in calculating aerosol/Rayleigh lookup tables and cloud masking. However, this baseline version assumes the values of the MODIS SRP, with no modifications for the GEO's very different observation of Earth's surface (regional versus global), and geometry (each site from a fixed sensor view, but widely varying solar angles). Because there are much larger differences in ground sampling and viewing geometry between LEO and GEO, the SRPs optimized for MODIS or similar LEO sensors do not appear be appropriate for GEO. For example, Gupta et al. (2019) finds biases in DT-algorithm applied to AHI on Himawari-8. Also, when analysing NOAA's aerosol product created from ABI, Zhang et al. (2020) shows the need for an empirical correction to improve mean bias and Root Mean Squared Error (RMSE).

Therefore, assuming that biases in retrievals may be due, at least in part, to GEO's sampling, we use strategies laid out by Levy et al. (2007b), and focus on GOES -16 ABI that is currently in the GOES-East position near -75°W (denoted as ABI East or ABIE). We derive the atmospherically corrected spectral reflectance (AC-ref) and investigate its angular variation and the variation of land cover type. The result is a new SRP, and we test this new SRP on ABI data and compare the aerosol results with the baseline (assumed MODIS SRP).

This manuscript is organized as follows. Section 2 introduces the original DT algorithm for MODIS and a baseline for GEO sensors. Section 3 compares GEO ABI and LEO MODIS observations from perspective of geometric differences and identifies an issue in the baseline AOD retrieval from GEO. Methodologies and input datasets for the atmospheric correction calculation

are described in Sect. 4, and an investigation of new SRP is conducted in Sect. 5. Section 6 presents the performance of the new SRP and compares the newly retrieved GEO AOD with AOD from Aerosol Robotic Network (AERONET). A discussion and conclusion are presented in Sect. 7 and Sect. 8, respectively.

## 2. Data and Methodology

### 2.1 MODIS Dark Target aerosol retrieval algorithm and products

Radiance from the sun is absorbed and reflected by constituents of the atmosphere and the surface. Observing from the Top-of-Atmosphere (TOA), a satellite sees the result of those interactions. For a theoretical scene that is both free of clouds and shadows, as well as free of trace gas absorptions, the TOA radiance is composed only of surface and aerosol/molecular scatterings.

MODIS, aboard the Terra and Aqua sun-synchronous polar-orbiting satellites, measures radiance of Earth-viewing scenes in 36 spectral bands spanning the deep blue at 0.41 μm to thermal infrared (TIR) at 14 μm. Each MODIS obtains near-global coverage twice a day, once during daytime and once at night at nominal spatial resolutions of 0.25 to 1.0 km, providing geolocated, calibrated spectral radiances known as the Level 1B data. The MODIS DT aerosol algorithm uses subsets of the spectral bands and follows separate logic to derive aerosol properties over land and the ocean.

For MODIS, we denote these bands as Blue, Green, Red, NIR1, NIR2, SWIR1 and SWIR2, centered respectively near 0.47, 0.55, 0.65, 0.86, 1.24, 1.64 and 2.11 μm. These are all in window regions with negligible or correctable trace gas absorption. The DT algorithm takes these L1B data, aggregates into N × N boxes of pixels, performs gas corrections along with cloud and other maskings, leading to an estimate of TOA cloud-free, gas-free reflectance in the 7 wavelength bands. Nominal resolution of this N × N box is 10 km. We denote this "vector" of multi-band reflectance as the "Mean_Reflectance", We use this Mean_Reflectance to perform the aerosol retrieval, and the results and the diagnostics collectively are contained in the Level 2 (L2) product. For MODIS, this L2 product is commonly known as MxD04 (with the x=O for MODIS on Terra and x=Y for MODIS on Aqua). The latest standard version the MODIS L2 product is archived as Collection 6.1 (C61).

Although both land and ocean retrievals are based on a Look Up Table (LUT) inversion approach, each uses its own set of assumptions, and appropriate Mean_Reflectance vector as input. This study focuses on the over-land part of the DT algorithm that uses Mean_Reflectance_Land vector. Details of the DT land algorithm are well described online (https://darktarget.gsfc.nasa.gov/) and previous DT algorithm studies (Levy et al., 2007b, 2010, 2013; Gupta et al., 2016; Remer et al., 2005), but we summarize here.

To retrieve the over-land aerosol characteristics, the signal from the atmosphere must be separated from the signal originating from the land surface beneath. The unique aspect of the DT algorithm lies in how the algorithm assumes the surface reflectance to make that separation – and the assumptions about the surface are known as the SRP. Based on low-flying aircraft measurements, Kaufman et al. (1997a) found that the value of surface reflectance in the Blue (e.g. 0.47 μm) and Red (0.65

μm) wavelengths were approximately ¼ and ½, respectively, of the surface reflectance in the SWIR2 (2.11 μm) over natural surface such as vegetation and dark soils. Physically, this relationship was expected based on the relative balance between absorption of visible radiation by chlorophyll versus absorption of SWIR2 radiation by the water within the vegetation [Kaufman et al., 1997b]. Since aerosol (especially fine-sized particles indicative of anthropogenic or burning processes) is often nearly transparent at 2.11μm, MODIS essentially observes the 2.11 μm surface reflectance. That initial estimate of 2.11 μm surface reflectance leads to an easy estimate of the 0.47 and 0.65 μm surface reflectance. This SRP (Blue = ¼ of SWIR2, Red = ½ of SWIR2), was coded into the at-launch MODIS algorithm (Kaufman et al., 1997a; Remer et al., 2005).

Although the initial MODIS aerosol product compared well to direct observations of AOD by sunphotometer (Chu et al., 2002; Remer et al., 2005), Levy et al. (2005) found some systematic biases that suggested revisiting SRP. The result was when regressing blue, red, and SWIR2 values, best-fit slopes were different from the original ratios, and there were non-zero y-intercepts. Furthermore, variability of the blue/red and red/SWIR y-offsets and slopes appeared to depend on observing geometry and surface type (Levy et al., 2007b). As a result, the current version of the MODIS-DT algorithm includes SRPs which also depend on scattering angle ($\Theta$) and surface "greenness" in the form of a Normalized Difference Vegetation Index (NDVI) based on NIR2 and SWIR2 channels (1.24 and 2.11 μm)

$$NDVI_{SWIR} = (\rho^m_{NIR2} - \rho^m_{SWIR2})/(\rho^m_{NIR2} + \rho^m_{SWIR2}), \tag{1}$$

where $\rho^m_{NIR2}$ and $\rho^m_{SWIR2}$ are the measured top-of-atmosphere reflectances in the NIR2 and SWIR2 wavelengths, respectively. The current version of the algorithm (e.g., MODIS Collection 6.1) uses SRPs as follows (writing SWIR instead of SWIR2 for simplicity):

$$\rho^s_{Red} = \rho^s_{SWIR} \times slope_{RedSWIR} + yint_{RedSWIR},$$
$$\rho^s_{Blue} = \rho^s_{Red} \times slope_{BlueRed} + yint_{BlueRed}, \tag{2}$$

where,

$$Slope_{RedSWIR} = slope^{NDVI_{SWIR}}_{RedSWIR} + 0.002\Theta - 0.27,$$
$$yint_{RedSWIR} = -0.00025\Theta + 0.033,$$
$$slope_{BlueRed} = 0.49,$$
$$and\ yint_{BlueRed} = 0.005, \tag{3}$$

where in turn

$$slope^{NDVI_{SWIR}}_{RedSWIR} = 0.58, if\ NDVI_{SWIR} < 0.25,$$
$$slope^{NDVI_{SWIR}}_{RedSWIR} = 0.48, if\ NDVI_{SWIR} > 0.75,$$
$$slope^{NDVI_{SWIR}}_{RedSWIR} = 0.58 - 0.2(NDVI_{SWIR} - 0.25), if\ 0.25 \leq NDVI_{SWIR} \leq 0.75. \tag{4}$$

The scattering angle ($\Theta$) is defined as

$$\Theta = cos^{-1}(-cos\theta_0 cos\theta + sin\theta_0 sin\theta cos\varphi), \tag{5}$$

where $\theta_0$, $\theta$, and $\varphi$ are the solar zenith angle (SZA), viewing zenith angle (VZA), and relative azimuth angles (difference between solar and sensor azimuth angles, RAA), respectively. Note that the NDVI_SWIR is calculated from TOA measured

(superscript "m") reflectance, but the SRP involve surface (superscript "s") reflectance. We can interpret the subscripts as "Red from SWIR' and "Blue from Red", respectively.

For Collection 6.1 of the MODIS product, Gupta et al., (2016) added a correction to the $slope_{RedSWIR}^{NDVI_{SWIR}}$ and $yint_{RedSWIR}^{NDVI_{SWIR}}$ to account for an urban surface. The urban correction takes into account the pixels with urban percentage (UP) larger than 20% as follows:

Where $NDVI_{SWIR} < 0.20$

$slope_{RedSWIR}^{NDVI_{SWIR}} = 0.78$ and $yint_{RedSWIR}^{NDVI_{SWIR}} = -0.02, if\ 20\ \% \leq UP < 50\%$ ,

$slope_{RedSWIR}^{NDVI_{SWIR}} = 0.66$ and $yint_{RedSWIR}^{NDVI_{SWIR}} = 0.02,\ if\ UP \geq 50\%,$         (6)

$slope_{BlueRed}^{NDVI_{SWIR}} = 0.51, if\ 20\ \% \leq UP < 50\%$

$slope_{BlueRed}^{NDVI_{SWIR}} = 0.52, if\ UP \geq 50\%$         (7)

and where $NDVI_{SWIR} \geq 0.20$

$slope_{RedSWIR}^{NDVI_{SWIR}} = 0.62$ and $yint_{RedSWIR}^{NDVI_{SWIR}} = 0.0, if\ 20\% \leq UP < 70\%,$

$slope_{RedSWIR}^{NDVI_{SWIR}} = 0.65$ and $yint_{RedSWIR}^{NDVI_{SWIR}} = 0.0,\ if\ UP \geq 70\%.$         (8)

$slope_{BlueRed}^{NDVI_{SWIR}} = 0.47, if\ 20\ \% \leq UP < 70\%$

$slope_{BlueRed}^{NDVI_{SWIR}} = 0.48, if\ UP \geq 70\%$         (9)

For MODIS, Urban Percentage is defined as the percentage of pixels (500m) identified as urban land cover type. The dataset of land cover type (MCD12Q1) will be introduced and described in Sect. 2.4.

This SRP is used as a constraint within the DT retrieval algorithm. The rest of the retrieval algorithm, including LUT calculation, gas correction, and cloud masking, aerosol model assumption and retrieval, is described in Levy et al. (2007a),

Levy et al. (2013), and Patadia et al. (2018).

The current assumptions for MODIS_SRP were derived by performing AC of the mean of the aggregated TOA Mean_Reflectance_Land vector (MODIS Level 2 MOD04/MYD04 products) over *globally* distributed AERONET sites. Even though we confirmed the basic SRP for use in Collection 6.1 (e.g., Levy et al., 2013), there have been both increases in AERONET coverage as well as a much larger dataset of Mean_Reflectance_Land. Yet, the ABIs are *regional* in coverage,

which means that only a subset of global AERONET sites can be observed by any single ABI. Therefore, to compare with the ABI datasets described in the next section, we perform AC on the subset of the AERONET sites that are observed by the corresponding ABI. We also constrain this analysis to the MODIS data (Aqua, MYD04 products) between 2015-2019. Single wavelength outputs from this exercise are known as AC-ref, which will be regressed to derive the SRP. Section 2.3 introduces AERONET data and a description of its collocation with ABI observations.

## 2.2 ABI and Baseline Dark Target aerosol retrieval algorithm

The ABI is a multi-band sensor aboard the GOES-R series of geostationary satellites. The GOES-R series currently in-orbit include GOES-16 (launched as GOES-R in November 2016) operating at the GOES-East position at 75° West longitude, GOES-17 (launched as GOES-S in March 2018) operating at the GOES-West position at 137° West until January 2023, and GOES-18 (launched as GOES-T in March 2022) that replaced GOES-17 as GOES-West in January 2023. Since 2019, all ABIs observe using a scanning pattern that results in "Full Disk" images every 10 minutes.

Each ABI has 16 channels, ranging from the blue (0.47 μm) to thermal infrared (13.3 μm), and vary in spatial resolution between 0.5 km, 1 km, and 2km (at subsatellite point). The Red channel (0.64 μm) is observed at 0.5 km (subsatellite point), while the Blue (0.47 μm), Near Infrared (NIR1; 0.86 μm), and SWIR1 (1.61 μm) bands at 1 km, with the remainder (including SWIR2; 2.24 μm) at 2 km. Note, ABI has neither a Green (~0.55 μm) nor 1.24 μm NIR2 band, in contrast to MODIS. As all ABIs include blue, red, and SWIR2 channels (0.47 μm, 0.64 μm and 2.24 μm) similar to MODIS, our initial assumption was that the MODIS SRP could be used for both ABIE (on GOES-East) and ABI-W (on GOES-West).

With that in mind, we are following the Gupta et al. (2019) approach for AHI and applying to ABI. The initial or "baseline" version of the DT aerosol algorithm on ABI generally follows the same logic as that on MODIS. the DT algorithm for ABI has key differences from that used for MODIS, including:

1) Aerosol/Rayleigh LUTs and gas corrections are pre-calculated for the ABI-specific wavelength bands, and aerosol types are assumed to have spectral refractive index with the same values as analogous wavelengths on MODIS (e.g., ABI → MODIS wavelengths: 0.47 → 0.47, 0.64 → 0.65, 2.24 → 2.11 μm). Note that the over-land retrieval does not require a green band reflectance as input, although it uses the same "indexed" AOD at 0.55 μm.

2) Using the observed L1B spectral reflectance and radiance data, the DT algorithm takes several steps for pre-processing and data aggregation. For both MODIS and ABI, we take N × N aggregations of native-resolution pixels to create box of 10 km. The difference is that ABI has one higher-resolution band. The common MODIS resolution is 0.5 km whereas GEO is 1.0 km. the ABI bands are aggregated by 10 x 10 pixels at 1 km to create a 10 km box, unlike the MODIS algorithm, which uses 20 x 20 pixels at 0.5 km to create the 10 km.

3) Cloud and ice/snow masking are modified to account for lack of 1.24 μm and some of the TIR bands. Details about the pixel masking for MODIS is described in DT ATBD (https://darktarget.gsfc.nasa.gov/atbd-land-algorithm) and Shi et al. (2021), and the modification for GEO is described in Gupta et al. (2019).

4) The NDVI$_{SWIR}$ test uses NIR1 (0.86 μm) rather than NIR2 (1.24 μm) for comparison with SWIR2 (2.24 μm) due to the lack of a 1.24 μm channel and the shift of this SWIR2 "2 μm channel" (from 2.12 to 2.24 μm). While not as "aerosol-free", vegetation reflects 0.86 μm similar to 1.24 μm (Miura et al., 1998). Let us denote NDVI$_{LEO\_SWIR}$ as NDVI$_{SWIR}$ defined in Eq. (1), and define NDVI$_{GEO\_SWIR}$ as;

$$NDVI_{GEO\_SWIR} = (\rho_{NIR1}^m - \rho_{SWIR2}^m)/(\rho_{NIR1}^m + \rho_{SWIR2}^m) \tag{10}$$

According to Karnieli et al. (2001) and Jin et al. (2021), both NDVI$_{SWIR}$ and NDVI$_{GEO\_SWIR}$ are well correlated with NDVI and

yet are less affected by atmospheric effects. We also conducted a study to check a consistency between NDVI$_{LEO\_SWIR}$ and NDVI$_{GEO\_SWIR}$ [Appendix A]. NDVI$_{GEO\_SWIR}$ matches NDVI$_{LEO\_SWIR}$ in dense vegetation where both NDVI$_{SWIRS}$ are high but falls to lower values at the low end in scenes with less vegetation coverage. Accordingly, in the SRP, the NDVI$_{GEO\_SWIR}$ increases the number of cases assigned into the low NDVI$_{SWIR}$ category (NDVI$_{SWIR}$ < 0.25) from 25.5% to 41.5%. This means that without modifying the threshold on NDVI$_{GEO\_SWIR}$ the SRP will be encountering pixels never used before by the DT

algorithm. The red surface reflectance will be parameterized differently in areas with less vegetation, which may lead to differences between DT GEO and DT LEO retrievals. It requires a new NDVI threshold or SRP improvement for the DT GEO retrieval.

For convenience, we denote as NDVI$_{SWIR}$ regardless of the difference in wavelength hereafter. We apply this baseline GEO DT algorithm to ABI observations to derive Level 2 Full Disk aerosol products at 10 km × 10 km nominal resolution, which

therefore includes its own "Mean_Reflectance_Land" vector (of TOA cloud-cleared, gas-corrected reflectances). Nominally, ABI produces six full disk images every hour in default Full Disk scan mode. However, to reduce data volume, we work here with one image per hour and limit to Full Disk data collected between July 2019 to June 2020. For ABI, one year is sufficient to produce the necessary statistics because of ABI's higher temporal resolution. Again, note that while the baseline DT-ABI algorithm uses the SRP defined by MODIS, AC of the Mean_Reflectance_Land is expected to lead to an improved definition

of SRP for ABI.

## 2.3 AERONET AOD and collocation criteria

The globally distributed AERONET network has provided aerosol optical properties for ~30 years (Giles et al., 2019), and has increased to nearly 540 active sites worldwide. The AERONET AOD dataset is widely used as ground truth for satellite retrievals because of its well-defined accuracy, instrument quality control, and strict regular calibration. The uncertainty of an

AOD measurement from a newly calibrated field instrument under cloud-free conditions is less than ±0.01 for wavelengths longer than 0.44 μm (Giles et al., 2019; Eck et al., 1999). Here, we utilize the Level 2.0 all-point, sun-observed AODs provided by the AERONET Version 3 algorithm and interpolate AERONET-provided spectral AODs to AOD at 0.55 μm using a quadratic fit in log-log space (Eck et al., 1999). The AERONET AOD at 0.55 μm is then used for two different purposes.

1) The AERONET AODs are used to generate an AC-ref from MODIS and ABIE observations. In this case, data are adopted where AOD at 0.55 μm is less than 0.2 and Ångström exponent (AE) (0.44 – 0.675 μm) is greater than 1.0. The AOD and AE is limited to avoid bias in AC-ref which can be caused by aerosol model assumption. Cases where coarse mode particle (AE<1.0) dominate are masked out to avoid large discrepancy in spectral AOD change between realistic and aerosol model assumption.

2)     The AERONET AODs are also used to validate the DT-derived AOD.

Spatiotemporal criteria for the satellite-AERONET co-locations are as follows. For AC-ref calculation, AERONET observations within ±15 minutes of satellite overpass (MODIS or ABIE) are collocated with satellite-derived TOA reflectance within ±0.3° rectangular grid centered over an AERONET site. Here, a relatively large spatial range was established to capture

TOA reflectance influenced by diverse land cover types and to mitigate potential cloud contamination. The AC-ref dataset consists of the spatial mean of TOA reflectance, spectral AOD from AERONET, land cover types present with the range, and observation geometries. Figure 1 displays number of data used for the AC-ref calculation at each AERONET site. For the purpose of AOD validation, a temporal criteria of ±15 minutes for AERONET AOD and a spatial criterion of ±0.2° rectangular grid for DT AOD are applied. The spatiotemporal collocation window follows standard DT validation practice used in the

MEaSUREs program.

## 2.4 Land cover type

MODIS Land Cover Type (MCD12Q1) version 6 products provide global land cover types from the Terra and Aqua combined measurement at yearly interval with 500 m sinusoidal grid resolution (https://lpdaac.usgs.gov/products/mcd12q1v006/). There are Land cover indices classified from six different classification scheme. In this study, International Geosphere-Biosphere

Programme (IGBP)'s classification (Belward et al., 1999; Sulla-Menashe and Friedl, 2018) is applied to investigate the changes in spectral relationship owing to land cover type. The IGBP's index classified 16 land cover types including Forests (index 1-5), Shrublands (index 6-7), Savannas (index 8-9), Grasslands (index 10), Permanent Wetlands (index 11), Croplands (index 12), Urban and Built-up lands (index 13), Cropland/Natural vegetation mosaics (index 14), Permanent Snow and ice (index 15), Barren (index 16), and Water bodies (index 17).

We first assign the percentage land type [%land type, hereafter] to the AC-ref derived at each AERONET site, using the same 0.3° distance around the AERONET station as done to calculate AC-ref. For the retrieval processing, 0.1° x 0.1° gridded map of land cover type is derived. Detailed collocation process is explained in Sect. 5.2.1 and Sect. 6.1.

## 3. Analysis of baseline DT-ABI algorithm

### 3.1 Differences in viewing geometries

MODIS (and its follow-on VIIRS sensors) provide global aerosol coverage, but by observing a given ground target at approximately the same time every day. To observe more rapid aerosol changes, as well as to characterize the aerosol diurnal cycle, we use imagers on GEO satellites. However, there are great differences between the observing geometry of ABI and MODIS. In a sun-synchronous polar-orbiting orbit, MODIS views a given ground target from a wide variety of VZA and RAA

over a period of several weeks, while SZA varies slowly. In contrast, ABI views each Earth scene from a constant VZA, while the sun moves from sunrise to sunset introducing a variable SZA. Figure 2(a) maps the VZA of ABIE. which is constant for all images. On the other hand, Fig. 2(b) and (c), show that the MODIS VZA varies, even on consecutive days. For example, when observing the GSFC AERONET site (red circle in Fig. 2), the VZA of ABIE is fixed at 45.42°, whereas the VZA of MODIS changes from 15.88° to 51.11°.

Figure 3 shows frequency distributions of scattering angle, SZA, VZA, and RAA of ABIE and MODIS (on Aqua) observations at the GSFC AERONET site (-76.84°E, 38.99°N). The data covers 5 years (2015-2019) of MODIS observation and a year (2019) of ABIE observation, respectively. In general, ABIE measures various solar angles as it provides multiple images for a location throughout a day. Figure 3a shows the SZA varying from 10° (noon during the summer solstice) to 90° (sunrise/sunsets all year), with the most frequent observations having SZA in the range of 60°-70°. With the sun moving from horizon to horizon during the day, the RAAs vary from 60° to 180° with the median value of 125.8° (Fig. 3c). This is from VZA fixed at 45.42° (Fig. 3e). Thus, under the fixed VZA condition, scattering angle of ABIE measurement changes in accordance with the variations of solar angles.

Compared to ABIE, MODIS measures limited solar angles because it flies in a sun-synchronous polar orbit. As both Terra or Aqua observe GSFC approximately ±1.5 hours from local solar noon, neither MODIS observes sunrise or sunset (SZA = 90°). Thus Fig. 3(b) shows that MODIS SZA is relatively evenly distributed between 20° and 70°, unlike ABIE SZA, which was peaked in the range of 60°-70°. The RAAs showed a bimodal distribution that peaked at 50° and 130° (Fig. 3d) and was absent in the range between 60° and 120°. Meanwhile, MODIS measures at various VZA conditions in the range of 0° to 60°. Accordingly, the scattering angle of MODIS is synchronized with the VZA variation rather than the limited solar angle variability. Although both scattering angle distribution in Fig. 3(g) and Fig. 3(h) cover the similar angular range, the main factor determining the variation is not the same.

Based on a comparison of observation geometries, it appears that LEO observation using the DT technique is more favourable to retrieve AOD than GEO observations. First, high SZA observations introduce greater uncertainty in AOD retrieval. Increased path length at high SZA makes it difficult to separate the aerosol contribution from other atmospheric components. The high SZA also can lead to uncertainty in AC-ref calculations and make the SRP analysis more difficult. Second, GEO observation are more likely to observe the 'vegetative hot-spot'. This is when the solar direction coincides with the observation direction ($\Theta \geq 175°$), resulting in a large increase surface reflectance at each wavelength. (Li et al., 2021) shows that ABI observe hot-spot frequently. The brighter surface overwhelms the aerosol contribution to the TOA reflectance and increases the uncertainty in AOD retrieval. Figure 3(g) also shows that ~4% of ABIE observations are performed at high scattering angles (>168°), whereas ~1% of MODIS observation reach that extreme at GSFC. The broad range of geometries lead to a difficulty in GEO AOD retrieval (Ceamanos et al., 2023) because the sensitivity to aerosols varies significantly during the day (Luffarelli and Govaerts, 2019; Ceamanos et al., 2019).

## 3.2 Bias in current ABI AOD and why we suspect surface reflectance.

We apply the baseline DT-ABI algorithm to ABI observations on GOES-16 (ABIE) and GOES-17 (ABI-West, ABIW) from August to September 2019 on an hour-by-hour basis. Figure 4 displays bias in the retrieved AOD versus AERONET AOD in each hour and SZA. Most obvious is that the DT AODs retrieved from both ABIE (Fig. 4a) and ABIW (Fig. 4b) show a time dependent bias variation and have a peak in the error near local noon. However the bias calculated from each sensor shows different time distributions. Figure 4(c,d) illustrates that the DT bias overall increases as SZA decreases, but with differences depending on the scattering direction between the Sun and the satellite. ABIE AOD exhibits a higher bias in the morning (Fig. 3c), as shown by the time dependence showcasing a distribution skewed towards the morning (Fig. 4a). In contrast, ABIW AOD shows a greater bias in the afternoon then the morning (Fig. 4d), aligning with the afternoon-skewed distribution (Fig. 4b). This time dependence is consistent in that the bias is relatively large when the direction in which the satellite faces the Earth's surface matches the direction in which sunlight arrives. In other words, the AOD bias is also influenced by the RAA, showing a reduced bias during low RAA conditions when the surface appears relatively dark due to shadows cast by the canopy.

We hypothesize that the diurnal signature in bias between each ABI and AERONET arises because the viewing geometry of the GEO sensor has different features than from the LEO one (Fig. 3). For GEO sensors, a particular ground site is always observed with a fixed VZA (Fig. 2(a) and Fig. 3(e)) while the sun angles change throughout the time. In contrast, from LEO sensors, a particular ground target will be observed from a variety of viewing zenith angles, while the solar zenith angle is relatively constant during a season. Since MODIS has a 16-day orbit repeat cycle (https://ladsweb.modaps.eosdis.nasa.gov/missions-and-measurements/modis/), any residual bias escaping the LEO SRP's compensation for anisotropic surface reflectance by assuming a dependency on scattering angle will be averaged out over MODIS's 16-day repeat cycle but be reinforced day after day with ABI.

From this point of view, we suspect that while assuming that the scattering angle represents the anisotropic reflectance pattern may work for MODIS on average, it would induce a large bias to GEO retrievals at local noon and/or dawn and dusk. This means we should consider a new SRP for ABI observations paying more attention to the details of the observational geometry than just the scattering angle. We must remember that from the view of the GEO sensor, each VZA matches up to a specific land cover type according to location. Differences in sensor specifications, such as wavelength shift and calibration status, may also be bring a necessity to reformulate the SRP. We will proceed with creating this new SRP and then show that it reduces biases and mitigates the bias's diurnal signature in retrieved AOD.

## 4. Atmospheric correction

### 4.1 Calculating Atmospherically-Corrected reflectance (AC-ref)

To develop a new SRP, we require a data set of spectral surface reflectance. To a first approximation, the wavelength-dependent reflectance measured by a sensor at the TOA reflectance $\rho_\lambda^*$ is the sum of contributions by the atmosphere-only (known as the path reflectance) and the surface-atmosphere interaction (Kaufman et al., 1997a).

$$\rho_\lambda^* = \rho_\lambda^a + \frac{F_{d\lambda} T_\lambda \rho_\lambda^S}{(1 - s_\lambda \rho_\lambda^S)}. \tag{11}$$

Here, the first term $\rho_\lambda^a$ is atmospheric path reflectance which consists of molecular and aerosol extinction, and the second term represent interaction of atmosphere and underlying surface. $F_{d\lambda}$ is normalized downward flux for zero surface reflectance, equivalent to the total downward transmission, and $T_\lambda$ is total upward transmission, $s_\lambda$ is atmospheric backscattering ratio, and $\rho_\lambda^S$ is surface reflectance. Note that for this equation, we assume that there is no 'extra' radiation arising from adjacent scenes (e.g., clouds) and that there is no absorption by trace gases. Also, note that this equation has been simplified for readability, although all terms have angular dependence. Given a radiative transfer model (RTM), knowledge of the aerosol type and loading, one can calculate all properties of the atmospheric contribution (e.g., $\rho_\lambda^a$, $F_{d\lambda}$, $T_\lambda$, and $s_\lambda$).

By rephrasing Eq. (11), surface reflectance, $\rho_\lambda^S$, can be written as shown in Eq. (12). If the $\rho_\lambda^*$ is the observation, and we somehow "know" the properties of the aerosol plus Rayleigh atmosphere (e.g., measured from AERONET), we can determine the $\rho_\lambda^S$. This process is known as AC, and the $\rho_\lambda^S$ derived from Eq. (12) is referred to as AC-ref.

$$\rho_\lambda^S = (\rho_\lambda^* - \rho_\lambda^a)/(s_\lambda(\rho_\lambda^* - \rho_\lambda^a) + F_{d\lambda} T_\lambda) \tag{12}$$

The AC-ref is an estimate of that surface reflectance, obtained by using an RTM to 'subtract' the atmospheric contribution from observed TOA values, given the knowledge of molecular scattering angle aerosol properties (Vermote et al., 1997). By calculating this surface reflectance in different wavelengths, we can determine the spectral relationships that we need for the DT aerosol retrieval algorithm. Let us focus on performing AC for ABIE only and compare with corresponding AC using MODIS observations over the ABIE domain. We assume that the TOA reflectance is the 10 km x 10 km Mean_Reflectance_Land parameter contained in the Level 2 aerosol product, which in turn corresponds to gas-absorption-corrected, cloud-masked and outlier-removed statistics of the original (Level 1B) spectral reflectance. The aerosol loading (e.g., spectral AOD) is observed by AERONET, and the RTM assumes the "Continental" (Remer et al., 2005) model to derive the spectral path reflectance and other atmospheric terms in Eq. (12). This process is summarized by the flowcharts in Fig. 5.

We need to atmospherically correct the blue, red and "2 μm" (2.11 μm for MODIS or 2.24 μm for ABIE) band to obtain surface reflectance at those wavelengths. AERONET measures AOD at several wavelengths in the visible so we calculated the 2nd order polynomial fit of spectral AOD between visible and SWIR channels. Interpolating AOD to the specific blue and red bands for MODIS or ABIE introduces minimal error. However, since there are no AERONET AOD measurements near 2 μm, we must extrapolate. Among the collocation dataset meeting the criteria outlined in Sect. 2.3, 84% includes valid AOD observation at 1.64 μm, so a second-order polynomial fit can be used to estimate the AOD at 2 μm with reasonable confidence.

The second order fit is also applied to the 16% cases with the longest observation wavelength of 1.02 μm. In this case, the extrapolation may induce greater error in the AOD estimation. Meanwhile, we use the generic Continental (Remer et al., 2005) model that provides spectral scattering and absorption properties, including a one-to-one relationship between extinction at all wavelengths in question. Although there are still uncertainties, by restricting the AC to situations where AOD at 0.55 μm is

less than 0.2 and AE is greater than 1.0, assuming Continental model, and extrapolating AERONET AOD where possible, we reduce the uncertainty related to aerosol model assumption for deriving AC-ref. In Appendix B, we tested the changes in the spectral ratio with respect to aerosol model assumption. The results show that different aerosol model assumptions lead to minor changes in the spectral relationship.

Figure 6 shows scatter plots of AC-ref of different wavelengths obtained from the GSFC AERONET site from a year of ABIE

observations. The AC-ref in the red wavelength (Fig. 6a) and blue wavelength (Fig. 6b) overall are strongly correlated with AC-ref in the SWIR and red, respectively, as currently assumed in the DT algorithm (Eq. 2). However, rather than being constant, regressions for both wavelength pairs change with SZA and correlate poorly when SZA is very high (> ~75°).

## 4.2 Testing the AC-ref to the DT algorithm

Here, we use the results in Fig 5, to test whether the SRPs help to improve the ABIE DT retrieval. Although the regression

quality varies with SZA, to a first approximation we see that both red/SWIR, and blue/red vary nearly monotonically with SZA.  We use this relationship to determine whether an SRP that includes SZA might improve the DT retrieval, at least at the GSFC site.  Figure 7 shows the results of this pilot study, using these new SRPs as compared with the AOD retrieved from the baseline SRPs at GSFC. Figure 7(a) shows correlations between the DT AODs and AERONET AODs, and Fig. 7(b) represents the biases in DT AODs for each hour. The baseline DT AOD correlates well with AERONET AOD with correlation coefficient

of 0.91 but is positively biased with slope of 1.44 and y-intercept of 0.01. Diurnal bias variation reveals a peak near noon time, like the comparison shown in Fig. 4(a).  This pilot study shows us that modification of SRPs varying with SZA mitigates the bias peak at noon (Fig. 7b). The new AOD shows a significant improvement in that 92.62% of retrieval falls within expected error (%EE) range (±0.05 + 15%) (Levy et al., 2013), compared to 50.82% of the baseline AOD falling within the %EE. While there is circularity in this test, with validation being done with the formulation data set, there are many steps between

formulation and validation and this test shows us (a) consistency through all those steps and (b) that changing the SRP assumption causes a response in the retrieval of the right order of magnitude to correct the bias.

## 5. Spectral Relation Parameterization

### 5.1 Comparison of SRP between MODIS and ABIE

To compare this study with the SRPs used now in the current MODIS algorithm and to derive new SRPs for ABIE we return
to the large data base of AC-ref, where at each collocation spectral AC-ref is used to calculate parameters of the spectral AC-ref relationships.

Figure 8(a) repeats the study of Levy et al. (2007b) with more recent data (2015-2019 and over the ABIE region only),
confirming the presence of non-zero offsets in the MODIS SRP. Even with a much-different sampling of AERONET data in
this (ABIE region only, different period) the overall red/SWIR and blue/red relationships are very similar to the earlier
regressions. Figure 8(b) applies the same AC-ref technique but for the ABIE observations. While the slopes and y-offsets are
overall similar to those observed when regressing MODIS, the visible and SWIR relationships are more scattered, with
significantly reduced correlation coefficient.

With MODIS, the variability in relationship between visible and SWIR AC-ref is controlled by parameterizing with
$NDVI_{SWIR\_LEO}$ and scattering angle as explained in Sect. 2.1. We repeat the entirety of the Levy et al. (2007b) study with the
recent MODIS data of the ABI FOV (Western Hemisphere) only, as well as with the ABIE data. The results are presented in
Fig. 9 and 10. The panels in Fig. 9 show changes of overall ratios (forced through zero) when separated into three bins of
$NDVI_{SWIR}$ ($NDVI_{LEO\_SWIR}$ or $NDVI_{GEO\_SWIR}$), whereas the panels in Fig. 10 show the values of regression (slope and y-intercept) for each of 20 bins of scattering angle. AC-ref are sorted according to scattering angle, binned into equal number
bins and regression parameters are calculated for each bin. The mean of each bin is plotted against a scattering angle.

For MODIS (e.g., Fig. 9(a), 9(c), 10(a) and 10(d)), the overall patterns remain similar to the equations shown in Sect. 2.1. In
Fig. 9(a), the slope decreases from 0.60 to 0.52 with an increase of $NDVI_{SWIR}$ from 0.25 to 0.50 like Eq. (4). The blue-red
slopes are almost independent of $NDVI_{SWIR}$ (Fig. 9b), as in the current DT MODIS algorithm (Eq. 4). For scattering angle, the
slope and y-intercept change (Fig. 10a and 9d, red) with rate of decrease/increase not much different from those in Eq. (3).
Note that the SRPs described by Eq. (3) also include a term dependent on $NDVI_{SWIR}$ that is not explicitly accounted for in the
Fig. 10 analysis, so exact dependencies on scattering angle are not expected to be identical. In Fig. 10(a), we see that indeed
the scattering angle dependence of parameters between blue and red is much weaker than their red-SWIR counterparts. In
conclusion, we find that the SRPs derived in this study from the MODIS AC-ref data base at AERONET stations in the western
hemisphere agrees within expectations with the dependence on scattering angle and $NDVI_{SWIR}$ being used in the current
MODIS DT algorithm that was derived years ago from a global database.

On the other hand, the spectral relationships of ABIE AC-ref are different from the MODIS ones. First, it is seen that the blue-red relationship has a variability that cannot be expressed as a constant. In Fig. 9(d), slope between blue and red is significantly
higher when the $NDVI_{SWIR}$ is greater than 0.5. We also find clear scattering angle dependence in the slope and y-intercept for
the blue-red AC-ref relationship from Fig. 10(b and e, blue). It can be presumed that the blue-red correlations show greater
scatter in Fig. 8(b) and Fig. 9(d) due to the changes of spectral relationship with viewing geometry.

While the change in slope between the red and SWIR AC-ref by NDVI$_{SWIR}$ (Fig. 9b) shows the similarity with the MODIS relationship, in Fig. 10(b), its dependency on scattering angle presents differently. Unlike the MODIS slope that increases linearly with the increase of the scattering angle, the ABIE slope shows a weak scattering angle dependence. In particular, when the scattering angle is higher than ~150°, the slope is significantly lower than that of MODIS. Since the angle dependence shown in Fig. 10(a) is sufficiently similar to the current DT assumption, comparing Fig. 10(a) and Fig. 10(b) shows that the

current SRP overestimates red slope and underestimates the blue slope when the scattering angle is high. This in turn leads to high bias in backscattered conditions near noon as shown in Fig. 4.

While the Red vs. SWIR regression shows weak dependence on the scattering angle, the regression changes linearly with SZA in Fig. 10(c) and Fig. 10(f). As SZA increases, the slope increases, and the interception decreases. The linear dependence is also evident in the blue-red relationship, although the points are slightly deviating from the linearity when SZA is high (>70°).

There are multiple possibilities for the large differences between ABIE and MODIS SRP parameters. One is that the wavelengths are different. MODIS blue, red, and SWIR channels are centered near 0.47 μm, 0.65 μm 2.11 μm, while corresponding ABIE channels are centered near 0.47, 0.64 and 2.24 μm, respectively. To test the impact of the wavelength shift, we used surface reflectance obtained from ASTER spectral library (Baldridge et al., 2009) that includes 2300 spectra of a wide variety of materials covering 0.35–2.5 μm with 0.001 nm resolution. We integrated bidirectional reflectance from 340

vegetation tree and 174 vegetation shrub cases for the specific wavelength pairs of each MODIS and ABI. The conclusion was that the wavelength shifts from MODIS to ABI results in negligible differences in red-SWIR relationship, while the blue-red slope decreases by 10% from 0.86 to 0.77 and the y-intercept increases from 0.001 to 0.003. These differences are not large enough to explain the differences between ABIE and MODIS SRP, as seen in Fig. 10.

The second and more likely reason for the differences is that ABIE and MODIS have very different viewing geometries, as

discussed in Sect. 3.1. While the scattering angle parameterization in the MODIS and baseline ABIE algorithm is meant to adjust the SRP parameters to account for anisotropic reflectance effects, spectral anisotropy is obviously more complex than can be modeled by a single parameter (e.g., Gatebe and King, 2016). ABIE's coupling of view angle with location has apparently accentuated the biases remaining from the one parameter formulation, while MODIS' mixing of view angles over each location has mitigated these biases. Note that Remer et al. (2001) find strong view angle dependence in observed ratios

of visible to SWIR, but that averaging over the range of view angles would bring observed ratios closer to the ¼ and ½ values expected at the time of their study. The Remer et al. (2001) study also found dependencies on land surface type and season.

## 5.2. Surface Reflectance Parameterizations for ABI-East.

### 5.2.1 Land Type

In Sect. 5, we discussed that, for ABIE observations, variability between visible and SWIR AC-ref (Fig. 8b) does not clearly depend on scattering angle. Spectral AC-ref relationships change with solar angles at a given location (e.g., Fig. 6) but appear

different at different sites. Therefore, we explore whether there may be other parameters to explain surface reflectance relationships. One possibility is to use a more explicit parameterization based on surface type.

The DT SRP attempts to account for land cover type (and seasonal changes) using the NDVI$_{SWIR}$ (Eq. 4) and scattering angle (Eq. 3) as a proxy for bidirectional reflectance. This appears to be sufficient for MODIS, where SZAs and vegetation conditions both co-vary on seasonal scales. Differences in day-to-day viewing geometry help to remove overall biases caused by the SRP. Nonetheless, there was enough remaining bias over non-uniform and isolated urban surfaces that the MODIS-DT retrieval added a correction based on urban percentage (Gupta et al., 2016).

The ABIE geometry presents a very different problem in that every ground location is isolated – there is no 'averaging' of viewing angle. Thus, unlike MODIS (or another LEO sensor), one may not be able to assume a generalized dependence on the scattering angle for all ground target locations. Of course, the pathological limit is that there is a unique BRDF and functional dependence of solar angle at every location viewed by ABIE. Developing this "map", however, would be extremely expensive (time and computationally), and would require similar efforts to develop these descriptions for ABIW, AHI and any future regionally observing geostationary imagers. Therefore, we attempt to simplify this problem by separating the globe into three canopy types (two vegetation types plus urban) that represent the darker surfaces used for DT retrieval. From there, we develop a three-tiered SRP and test whether that can be used for ABIE and other GEO sensors.

According to the IGBP index, we classify surfaces dominated by deciduous or evergreen Forest (IGBP index 1 ~ 4) as Closed Vegetation (CV) and other vegetation (IGBP index 5~10, 12, 14) as Open Vegetation (OV). Figure 11 shows the global map of IGBP index and percentage of each land cover type (%land type) at 0.1 x 0.1° resolution, which are obtained from the MCD12Q1 products at 500 m resolution. By counting individual IGBP indices from all sub grid pixels, we obtain the most frequent IGBP index (Fig. 11a) and percentage of CV (Fig. 11b), OV (Fig. 11c) and urban (Fig. 11d) in a grid box. OV dominates over most of the global land mass. CV is dominant in 17.6% of non-water, ice-free, and non-barren pixels, and is generally concentrated near the equator. Meanwhile, the urban is dominant in 0.7% of the situations and is sporadically distributed.

Figure 11 shows that multiple land types can exist simultaneously in a grid box in most cases where the surface properties are not uniform over a wide range. Since AC-ref is based on the contribution from each type, it can be assumed that the spectral AC-ref relationship is dependent on the %land type of the area for which the AC-ref is calculated, just as the urban percentage changes the SRP in the current algorithm. We intend to create an SRP for each land type, Open and Closed Vegetation, in the same manner that urban percentage is used in the current DT algorithm. We will do this in two steps. First, we develop the land type's SRP. Second, we apply the new SRP to the retrieval.

The SRP analysis clusters the AC-ref dataset into three land types. Each cluster integrates cases where the land type occupies more than 10%, allowing us to observe changes in SRP corresponding to alterations in the percentage of each type. In the retrieval process, the algorithm now assumes a homogenous surface, even when the surface is heterogeneous. The algorithm reads the %land type maps (Fig. 11) and identifies the highest percentage land type in each pixel. The algorithm uses open vegetation by default and adjusts the SRP where the %CV or %UP is higher than the %OV. For example, if we assume a grid

box is covered by 20% urban, 50%CV, and 30%OV case, the algorithm chooses CV and calculates the SRP for 50% CV by using the multiple linear regression described in Sect. 5.2.2. The contribution of the other 50% of the pixel is not considered here. Accordingly, the following unfortunate case will be possible. If the pixel is filled with 40%CV, 30%OV, and 30%urban, the pixel is identified as CV even though 60% of pixel is covered by other types. Ideally, a weighting function that takes into account significant percentages of all surface categories would be preferable and such a weighting function will be investigated in a subsequent study. This section describes the process of creating the SRP's specific to each land type, leaving further description of the application of SRP to Sect. 6.1.

In Fig. 12, we plot the Red and SWIR AC-ref relationship by land type. A population of AC-ref shown by gray dots were categorized into three group as described before, and a linear regression was fitted for each AC-refs group. AC-ref in each group were binned into 10 equal bins, and the mean and standard deviation of each bin was calculated and shown by colored circles and vertical bars. The CV is dark at both SWIR and red wavelengths and has a lower slope compared to the OV and urban. Meanwhile, for urban, red vs. SWIR AC-ref is higher than the other land surface types, causing a higher slope than the other two groups. Since OV encompasses all the various land cover types that are not classified as CV or Urban and dominates the statistics, the regression of this type is similar to the overall correlation where no surface type is specified, as shown in Fig. 8 (b, red). In Fig. 13 we look for dependencies of the regression by plotting the slope against SZA, $NDVI_{SWIR}$ and the different land cover type percentages. In this figure, we simplified the regression to force the y-intercept to 0 and display the derived slope to make it easier to see the linearity of the regression change with the parameters. For vegetative surface, slopes from the red-SWIR regression range between 0.4 to 0.6, and slope variability is well captured by parameterization using $NDVI_{SWIR}$. The slope of the CV group shows a weak dependence on the parameters, with a slight decrease observed with increasing $NDVI_{SWIR}$ (Fig. 13b, red) and %CV (Fig. 13c, red). When the surface becomes even darker, where $NDVI_{SWIR}$ is greater than ~0.6, the regression tends to lose its sensitivity to the $NDVI_{SWIR}$. The slopes of the OV group behave similarly, but the dependence on $NDVI_{SWIR}$ and %OV is stronger compared to the CV group. An increase in vegetation percentage is expected correspond to a darkening of visible surface reflectance, and thereby decreases the Red vs. SWIR slope. The urban slope also decreases as $NDVI_{SWIR}$ increases, but as % urban increases, it increases due to the greater contribution of high reflectance from human-made structure. The slope is high when SZA is low for groups of OV and urban, but in the CV group, a weak positive change is shown.

### 5.2.2 Multiple Linear Regression for Red and SWIR relationship

Previously, we discussed the inability to parameterize the angular dependence of the SRP with only scattering angle in the same way as is done with the current MODIS DT algorithm (Fig. 10). We also discussed that the correlation between visible and SWIR AC-ref obtained from ABIE varies with land cover type (Fig. 12) and investigated a dependency on $NDVI_{SWIR}$ and %land type (Fig. 13). Based on the above analysis, for the red and SWIR relationship, we introduce a new SRP that takes into account SZA, $NDVI_{SWIR}$ and %land types through multiple linear regression. In this way, anisotropic reflectance is

parameterized as a function of SZA as before, but that relationship is modified simultaneously by NDVI$_{SWIR}$ and %land. Also, the urban correction is no longer being performed as an add-on process as it is in the current MODIS DT algorithm.

Like the current MODIS DT algorithm, this study aims to parameterize the regression coefficients of spectral relationships of surface reflectance. We predict the slope and y-intercept as a function of input parameters and then derive the red surface reflectance from the SWIR surface reflectance based on those coefficients. Table 1 summarizes the multiple linear regression (MLR) between input parameters and the coefficients for each land type. Each MLR coefficient listed in the table represents the change in predicted value per unit change in the predictor, holding all other input variables constant. The constant is equal

to the predicted value when all the input parameters are zero. That is, according to the MLR coefficient, the key factor to change the spectral relationship is NDVI$_{SWIR}$ for OV and urban, but it can be seen that %land type plays an important role in CV and urban conditions. Note that while the MLR coefficients for SZA and %land type seem small, the value of SZA and %land type ranges from 0 to 70 and from 0 to 100, respectively, whereas the value of NDVI$_{SWIR}$ ranges from 0 to 1. Multiplying MLR coefficients by typical values can place the influence of SZA on the same scale as the land parameters.

Taking an OV-dominated pixel as an example, based on the MLR coefficients provided in Table 1, the slope and y-intercept can be predicted using the following equations:

$$Slope^{OV}_{RedSWIR} = 0.0014 \times SZA - 0.4477 \times NDVI_{SWIR} + 0.0001 \times \%OV + 0.6729 \qquad (13)$$

$$Slope^{OV}_{RedSWIR} = -0.0003 \times SZA + 0.0411 \times NDVI_{SWIR} - 0.0001 \times \%OV - 0.0041 \qquad (14)$$

For a given NDVI$_{SWIR}$ and %OV, the MLR yields high Red-to-SWIR ratio when SZA is low.

**5.2.3 New SRP for Blue and Red relationship**

The Blue and Red relationship does not change significantly with land type. We account only for the NDVI$_{SWIR}$ dependence and angular change which are shown in Fig. 9 and Fig. 10. The blue and red relationship is parameterized as a linear function of scattering angle for three NDVI$_{SWIR}$ groups as shown in Fig. 14.

In Fig. 14, the ratio of blue to red is close to 0.6 under backscattering conditions and is not significantly distinguished according

to NDVI$_{SWIR}$. The blue-red relationship of the three NDVI$_{SWIR}$ groups is differentiated in terms of dependence on the scattering angle. The slope decreases slightly as the scattering angle decreases when NDVI$_{SWIR}$ is lower than 0.5, whereas when NDVI$_{SWIR}$ is high, the transition from backward to forward scatter increases the slope. It is seen that the ratio is as high as 1 under forward scattering condition. The high NDVI$_{SWIR}$ mostly corresponds to the dense vegetation such as tropical forests and crops at their peak growth status. The forest canopy induces shadow-driven reflectance in forward scattering, which

darkens blue and red reflectance and consequently a 1 to 1 regression between them.

## 6. Result

### 6.1. Applying the new SRP in the DT algorithm

The newly developed SRPs are applied to the DT algorithm for ABIE, and performance is tested in terms of predicted surface reflectance and retrieved AOD. The visible surface reflectances predicted from the AC-ref at the SWIR wavelength obtained from ABIE observation are compared to the visible AC-ref in Sect. 6.2, and the AODs retrieved from the modified DT algorithm for ABIE by adopting the new SRPs are validated with AERONET AOD in Sect. 6.3.

The ABIE-DT algorithm follows the same flow as described in Sect. 2.1 but assumes the new SRP for both red and blue surface reflectance estimation. As different SRP assumptions are made depending on the land cover type, a classification of land type is performed based on an ancillary map of %CV, %OV and %urban before estimating the surface reflectance. Considering the 10 km resolution of DT retrieval, we applied the map of %land type derived from a 0.1 x 0.1° grid box in the same manner as we used to produce the maps in Fig. 11. The classification process finds the %CV, %OV and %urban from the nearest grid box for each location, then assigns the pixel to be CV or urban if the %CV or %urban is higher than others. In case of equal percentage of two different type in a grid, we put priority in order of urban, CV, and OV. For example, if both %CV and %urban occupies 40% of the area respectively, the pixel is assigned as urban. If none of conditions are met, the DT retrieval assumes OV as the default.

### 6.2. Comparison of predicted surface reflectance

Figure 15 shows the comparison between the predicted surface reflectance and AC-ref in blue and red, respectively. It also compares the performance of the new SRP to that of the baseline SRP applied to the MODIS DT algorithm. In Fig. 15(a), it is seen that the baseline SRP overestimates the red surface reflectance in CV but underestimates it in OV. In urban, the predicted surface reflectance correlates closely with AC-ref, but with a higher root-mean-square error (RMSE) than other land types. The surface reflectance predicted by the new SRP (Fig. 15b) represents a smaller RMSE in CV and urban compared to the baseline product. It is also bringing the predicted surface reflectances closer to the atmospherically corrected values. When it comes to prediction of blue surface reflectance, both the baseline and new SRP produce estimated surface reflectances that underestimate AC-ref, but the new SRP lowers the RMSE relative to the original values for all land surface types.

There is a degree of circularity in comparing the resulting estimated surface reflectances to the same data from which they were derived. However, many factors come into play during the derivation and there is no guarantee that estimated reflectances will match the AC-ref any better than those derived from the baseline SRP. However, the first basic step is to prove that the estimated reflectances do indeed match the statistics of their formulation data set. As such the results presented in this section are a necessary but not comprehensive proof of the new parameterization.

## 6.3. Comparison of DT-GEO AOD and AERONET AOD

The goal of the new SRP is not to derive surface reflectance, but to improve the AOD retrieval from GEO observations. Therefore, we use the new SRP in the DT algorithm applied to ABIE, which provided the formulation data set, but also its sister sensor ABIW, which remained independent of the derivation.

Figure 16 shows the new GEO DT AOD retrieval from ABIE on September 6[th], 2019, with 3-hour interval between 13 and 22 UTC. An aerosol plume crossing from north to south over Missouri is captured from the retrieval (Fig. 16. A-d), and other small plumes are detected around Houston and Louisiana as well. We select two AERONET sites adjacent to the aerosol path and compare the new AOD with the baseline AOD and AERONET AOD (Fig. 16e and f). At NEON_KONZ [39.10°N, -96.56°E], both the baseline and the new AOD follows the decreasing AERONET AOD between 14 and 23 UTC, albeit with a positive bias. The AOD at IMPROVE_MammothCave [37.13°N, -86.15°E] is as low as 0.1 at 13 UTC, but consistently increases as the aerosol plume approaches and peak at 23 UTC. Comparing the DT AODs, the new AODs are mostly lower than the baseline AODs, especially during local noon hours (17~19 UTC).

The initial issue with the DT algorithm applied to ABI sensors data was shown in Fig. 4 where we noticed an overall high bias that became maximum at around noon. Figure 17 illustrates the mitigation of the diurnal signature of the high AOD bias when applying the new SRP. The red dots recreate the original diurnal pattern from Fig. 4, and the black dots are the result of the new SRPs. The bias from AERONET of the new retrieved AOD is lower and less time dependent than the original. The new SRP alleviates the AOD bias when SZA is low (Fig. 4c vs. Fig. 17c) but shows less improvement at dusk/dawn. Although the new SRP was developed using AC-refs obtained from ABIE, applying the new SPR to the ABIW DT retrieval also mitigates the high AOD bias around noon as shown in Fig. 17(b). Thus, the ABIW results provide independent validation of the success of the new SRPs.

Figure 18 displays the average bias of the retrieved AOD against AERONET at each AERONET site. Figure 18(a and b) are side by side comparisons of the results of the old and new SRPs, respectively for ABIE. The products of ABIW are compared in Fig. 18(c and d). The ABIE AOD retrieved from the MODIS DT algorithm (Fig. 18a) agree with AERONET AOD with differences ranging between 0.0~0.2 at most stations. The positive bias is also prominent in ABIW AOD across western North America. Figure 18(b and c) show that the retrievals using the new SRP decrease the bias in both ABIE and ABIW AOD. However, the GEO-DT algorithm still overestimates AOD across most of western North America and some random locations. At some locations across the region, a positive bias turns into a negative one, but not below -0.05. The overall picture is one of improvement with stations closest to the subsatellite point with the smallest SZA at noon seeing the largest success.

The scatter plots in Fig. 19 compares the DT AODs with AERONET AOD. By applying the new SRP, the %EE has been improved from 50.53% to 75.57%, and the regression slope became closer to 1. Figure 20 shows the same plots as seen in Figure 17 but for three specific AERONET site, GSFC, NEON Harvard, and PNNL, which are dominated by Urban (44.09%), mixed forests (47.57%), and Croplands (86.87%), respectively. The AERONET sites are observed from ABIE with fixed VZA of 45.42°, 49.38°, and 68.99°, respectively. The new SRP reduces the AOD bias at the three AEROENT site but has less

impact on the mitigation of the time-dependence at PNNL. The AOD retrieved at PNNL may have greater uncertainty in that the site is located at the high VZA near the boundary of ABIE's observing disk, and the cropland surface contributed to the bright reflectance.

## 7. Discussion

The DT algorithm does not require building a database of surface reflectance using several years of satellite observations and thus is quickly adaptable to new sensors. This is because it constrains surface reflectance dynamically in real-time. The constraint is based on a physical connection between light absorption in the visible and SWIR in plant leaves and bare soil. The physical connection leads to empirically derived spectral relationships between surface reflectance in different bands. The surface reflectance parameterization (SRP) approach can be valid for any sensor. However, differences in viewing geometries between LEO and GEO sensors introduced an issue in the SRP parameterization. We explore this issue by calculating surface reflectance from top-of-atmosphere reflectance measured by a LEO sensor (MODIS) and a GEO sensor (ABIE) using atmospheric correction at AERONET sites.

In the MODIS case, using the most recent 5 years of observations confirmed the same SRPs and the same dependency on scattering angle established 20 years ago. However, an assessment of ABIE AC-ref determined that the GEO SRPs were inconsistent with the MODIS-based SRPs. At the outset these differences could be attributed to either spectral differences in the wavelength bands of each instrument or to the viewing geometry differences between a LEO and GEO. A sensitivity study proved that spectral differences cannot explain the magnitude of the differences that we are seeing. However, a remarkable difference between the SRPs of MODIS and ABIE was seen in the dependence on the scattering angle, which suggests geometrical differences play a significant role. The inconsistency increases bias in visible surface reflectance when the scattering angle is higher than 150°. We note that for ABIE, the relative azimuth angle (RAA) range at each site spans 40° to 180°. Meanwhile, the MODIS RAAs are limited to two sectors: 40° to 70° and 120° to 150°. For MODIS, within each sector the variety of VZA encountered dilutes geometrical differences for average values. Most importantly MODIS never measures RAA > 150°, near the vegetative hotspot.

The key to the differences between MODIS and ABIE is based on the fact that GEO and LEO measurements have different viewing geometries, and changes in scattering angles are driven by different factors in each sensor. In ABIE, solar angle varies, but VZA stays constant at each location, while MODIS measures a narrow range for solar angle while observing a wide range of VZA. This has two implications. First, for ABIE, because VZA is constant at each site, geometry is convolved with surface characteristics such as land cover. Thus, a dependence of SRP on the scattering angle in the ABIE analysis may be a proxy for land cover type. Second, ABIE encounters different combinations of VZA, SZA, and RAA that MODIS never does. The baseline parameterization appears to continue to serve DT MODIS well, suggesting that the decreased ability of the baseline parameterization to serve ABIE may lie in the new geometrical combinations that are now appearing in ABIE.

Although MODIS DT SRP accounts for angle-dependent bidirectional reflectivity, it is imperfect for modelling anisotropy for a variety of condition. The connection to the vegetative hot spot as the source of the differences in the spectral relationships of MODIS and ABIE suggests that ignoring surface cover characteristics could be the root cause of lingering uncertainties in the MODIS DT SRP and the reason for ABIE's poor performance. One way to illustrate the convolution between geometry and surface properties in the ABIE data is to examine in detail the situation in differing land covers. Fig. 12 and 13 demonstrate that AC-refs classified into Open Vegetation (OV), Closed Vegetation (CV), and Urban have different spectral relationships. The regression coefficients between red and SWIR AC-ref vary with land cover type and show dependence according to the homogeneity of land cover and $NDVI_{SWIR}$. The blue and red spectral relationship does not differ significantly depending on the land cover, but unlike the DT MODIS relationship, the variability according to the $NDVI_{SWIR}$ is evident.

We note that even the LEO DT algorithm began to explicitly address land cover type in its SRP when that land cover was Urban beginning with Collection 6.1 but continued to group all vegetation types together. However, with GEO, the novel geometry that includes the vegetation hotspot and convolves VZA with a specific location requires separate SRP for at least two vegetation categories. In this study, an improvement of SRP was achieved by applying the %land type parameterization, which distinguishes between open and closed canopy. However, it still does not fully characterize the surface BRDF, potentially resulting in residual angular bias in the surface reflectance and AOD. To ensure consistent DT retrievals across both LEO and GEO platforms, it is advisable to conduct a full characterization of the surface BRDF using a combined GEO-LEO AC-ref dataset. This dataset should cover a range of VZA, SZA, and RAA for each specific location.

The approach to calculating %land type also requires improvements in that there is a discrepancy between map of %land type and the ground pixels observed from GEO sensor. Concerns about spatial discontinuity in the AOD retrieval according to categorization of land cover type need to be addressed as well.

We acknowledge the need for further research to improve AC-ref accuracy. This should address concerns related to spatial resolution, uncertainties in high SZA conditions, and assumptions regarding aerosol models.

## 8. Conclusion

The new GEO sensors: Advanced Himawari Imager (AHI) on the Himawari satellite, the Advanced Baseline Imagers (ABIs) on the GOES- East and GOES-West sensors, the Advanced Meteorological Imager (AMI) on the Geostationary Korea Multi-Purpose Satellite-2A offer the aerosol community unprecedented opportunity to explore the temporal characteristics of aerosol with applications for air quality monitoring, resolving a developing smoke plume near its fire source, the co-evolution of aerosols and convective clouds on time scales of the convection and other possibilities. However, the same robust aerosol product that the community has come to expect from MODIS and VIIRS must be produced from the GEO sensors. Most importantly, any biases found in the retrieved AOD must be diurnally constant otherwise the very phenomena of interest to the community can be aliased by the diurnal signature in the error of the product.

We began this study with a baseline version of the DT-algorithm for GEO sensors including ABI. This includes lookup tables that account for shifted wavelengths, and modifications to cloud masking due to different resolutions and missing wavelengths. Based on our experience with porting to VIIRS, we assumed that the SRPs used for MODIS would be appropriate for the GEO sensors. Previous to this study, initial evaluation of the baseline GEO algorithm applied to AHI showed a correlation with collocated AERONET AOD and mean bias and RMSE only a little less accurate than MODIS AOD at the same stations and

time period (Gupta et al., 2019). These were encouraging results that propelled the baseline GEO algorithm to be applied to ABIE and ABIW with the same SRPs and for the resulting ABI AOD to be included in a global product consisting of AOD derived from three GEO sensors and three LEO sensors (Gupta et al., personal communication). Initial validation of ABIE and ABIW, as part of this merged product, show overall validation statistics comparable to MODIS, especially in terms of the percentage within expected error (%EE), but with a higher bias (Gupta et al., personal communication).

In this study we specifically examined the diurnal signature of differences with collocated AERONET AOD and found a distinctive diurnal signature of the bias with an amplitude of 0.10. The work done in the present study has focused on reducing the diurnal signature of the AOD bias against AERONET, which will strengthen the applicability of the DT ABI products for characterization of aerosol diurnal properties. While the baseline algorithm produces a robust and reasonable global product, by developing a new SRP from ABIE atmospherically corrected reflectances we may be able to cut the overall positive biases

in half to 0.01 to 0.08 and more importantly flatten the amplitude of the diurnal signature to less than 0.05. The new SRP parameterization was applied to ABIE measurements, the same data set that was used for its formulation, but it was also tested with ABIW measurements that provide an independent verification. The ABIE results are slightly better than ABIW results, as expected, but the independent ABIW results conform to the same error bars described above: a positive bias of less than 0.08 and a flattened diurnal signature with amplitude less than 0.05. We have not applied this new SRP to AHI or other GEO

satellite sensors. Because AHI views a different part of the globe with different vegetation, topography and soil types than do the ABI sensors, this specific parameterization fine-tuned for the Americas may not work as well for AHI's view of Asia and Oceania. With all their temporal promise, GEO sensors are essentially regional instruments by definition. It is possible that to maximize capability of each GEO sensor individual parameterizations may be necessary and regional versions of the DT algorithm more appropriate. A balance must be found between regional tuning and the goal of maintaining global consistency

so as not to lose the ability to characterize the global aerosol system as a whole.

    The investigation of the validity of applying the SRPs developed here for AHI and the question of balancing regional tuning with a global perspective are beyond the scope of this study. For now, we have a robust algorithm for the over land retrieval of aerosol that can be applied to the ABI instruments. The algorithm's parameterizations are based on understood physics of spectral light absorption and scattering by vegetative canopies. The parameterization described here will be implemented into

the DT package, available ABI observations will be processed, and the product made publicly available.

**Appendix A. NDVI$_{\text{LEO\_SWIR}}$ vs. NDVI$_{\text{LEO\_SWIR}}$**

We conducted a study to check a consistency between NDVI$_{\text{LEO\_SWIR}}$ and NDVI$_{\text{GEO\_SWIR}}$. The study is based on the MODIS TOA reflectance, which has both 0.86 μm and 1.24 μm channel. We derived the indices for 3.5 years (01.2017-06.2020) at six selected AERONET sites (Table A1) covering different land cover types.

The Fig. A1 shows NDVI$_{\text{LEO\_SWIR}}$ and NDVI$_{\text{GEO\_SWIR}}$ has a good consistency with linear regression as follows.

$$\text{NDVI}_{\text{GEO\_SWIR}} = 1.30 \times \text{NDVI}_{\text{LEO\_SWIR}} -0.20 \tag{A1}$$

**Appendix B. AC-ref change with aerosol model assumption.**

In order to test the impact of aerosol model assumption, we compared the AC-refs derived assuming different aerosol model. There are two groups of AC-ref obtained at GSFC AERONET site. The first AC-ref group is the same as the one described in Sect. 4.1. It was derived by feeding the TOA reflectance and AOD to the LUT inversion by assuming the Continental model. The second group is derived by using the 6sV simulation. We provided the TOA reflectance, AOD, Particle Size Distribution (PSD), and Refractive Index (RI) into the 6sV atmospheric correction process, and obtained the AC-ref. For the process, we first integrated the ABIE TOA reflectance, AERONET direct AODs, and AERONET inversion parameters (PSD and RIs), which are obtained from the GSFC AERONET site between July 2019 and June 2020. The study employs an inversion product with a hybrid scanning scenario, ensuring accurate parameters under following conditions (Sinyuk et al., 2020):

[1]. Single Scattering Albedo (SSA) and refractive index: 440 nm AOD larger than 0.4 and SZA larger than 25°.

[2]. Particle size distribution: 440 nm AOD larger than 0.02 and SZA larger than 25°.

Due to the accuracy criteria and less measurements frequency, the AERONET inversion products are available less frequently than the AERONET direct measurement, and retrievals of refractive index cannot be expected at low AOD which meet the condition of AC-ref calculation. Therefore, we allowed a wide time window (±1 Hours) for the collocation with an assumption that aerosol model does not change drastically within two hours.

Here, we have two groups of AC-ref which are derived by assuming a fixed continental model and a near real time aerosol model. Then we analysed the Visible vs. SWIR relationships as shown in the Fig. B1. In Fig. B1, the regression slopes (with forced zero y-intercept) of Red vs. SWIR and Blue vs. Red AC-ref pairs are plotted as a function of scattering angle. The symbol represents the aerosol model assumption (square: continental model, circle: AERONET inversion-based model), while the colour represents wavelength pair (red: Red vs. SWIR, blue: Blue vs. Red). For both wavelength pair, the aerosol model assumption does not make a meaningful change in terms of the value of slope and its dependence on scattering angle. Note that the result can be different where the location impacted by other aerosol types such as biomass burning.

## Data availability

The Dark Target (DT) algorithm has been ported to ABIs and produces aerosol data as part of the NASA MEaSUREs project (ROSES-2017; https://www.earthdata.nasa.gov/esds/competitive-programs/measures/leo-geo-synergy). The resulting products are accessible through https://ladsweb.modaps.eosdis.nasa.gov/archive/allData/5019. The NOAA GOES-R Series Advanced Baseline Imager (ABI) Level 1b Radiances can be obtained via doi:10.7289/V5BV7DSR (NOAA NCEI; GOES-R Calibration Working Group and GOES-R Series Program, 2017). The AERONET direct sun measurement data used in this study are available via the AERONET website (https://aeronet.gsfc.nasa.gov/new_web/download_all_v3_aod.html, AERONET team, 2023; Giles et al., 2019). MODIS Land Cover Type (MCD12Q1) version 6 products (DOI: 10.5067/MODIS/MCD12Q1.061) are accessible through LAADS DAAC (https://ladsweb.modaps.eosdis.nasa.gov/search/order/1/MCD12Q1--61, last accessed: January 2024).

## Author contributions

MK, RL, and LR designed this study. MK and SM developed retrieval codes and carried the experiments; MK, RL, and LR analyzed the data and result. PG guided land type percentage calculation. MK prepared the manuscript draft, and it was reviewed by all authors.

## Competing interests

The contact author has declared that none of the authors has any competing interests.

## Acknowledgements

This research was supported by the NASA Postdoctoral Program (NPP) at NASA Goddard Space Flight Center administered by Oak Ridge Associated Universities under contract with NASA. Furthermore, research support was extended by the NASA MEaSUREs project [NNH17ZDA001N-MEASURES: Making Earth System Data Records for Use in Research Environments].

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

**Table 1 Summary of new surface reflectance parameterization for DT-GEO algorithm. Multiple linear regression, which consists of three parameters, SZA, NDVI$_{SWIR}$, and % land type, predicts regression slope and y-intercept between Red and SWIR surface reflectance. The listed value refers the regression coefficient and the number in parentheses indicates 1-sigma of the coefficients.**

| Closed Vegetation | SZA | NDVI$_{SWIR}$ | % land type | Const |
|---|---|---|---|---|
| Slope (1-$\sigma$) | 0.0015 (0.0006) | 0.0181 (0.0540) | -0.0013 (0.0003) | 0.4439 |
| Y-int (1-$\sigma$) | -0.0002 (0.0001) | -0.0125 (0.0062) | 0.0000 (0.0000) | 0.0172 |
| Open Vegetation | SZA | NDVI$_{SWIR}$ | % land type | Const |
| Slope (1-$\sigma$) | 0.0014 (0.0005) | -0.4477 (0.0563) | 0.0001 (0.0003) | 0.6729 |
| Y-int (1-$\sigma$) | -0.0003 (0.0001) | 0.0411 (0.0084) | -0.0001 (0.0003) | -0.0041 |
| Urban | SZA | NDVI$_{SWIR}$ | % land type | Const |
| Slope (1-$\sigma$) | 0.0013 (0.0006) | -0.3890 (0.0620) | 0.0018 (0.0005) | 0.5976 |
| Y-int (1-$\sigma$) | -0.0003 (0.0001) | 0.0369 (0.0091) | -0.0001 (0.0001) | 0.0024 |


AERONET site used for AC_ref calculation

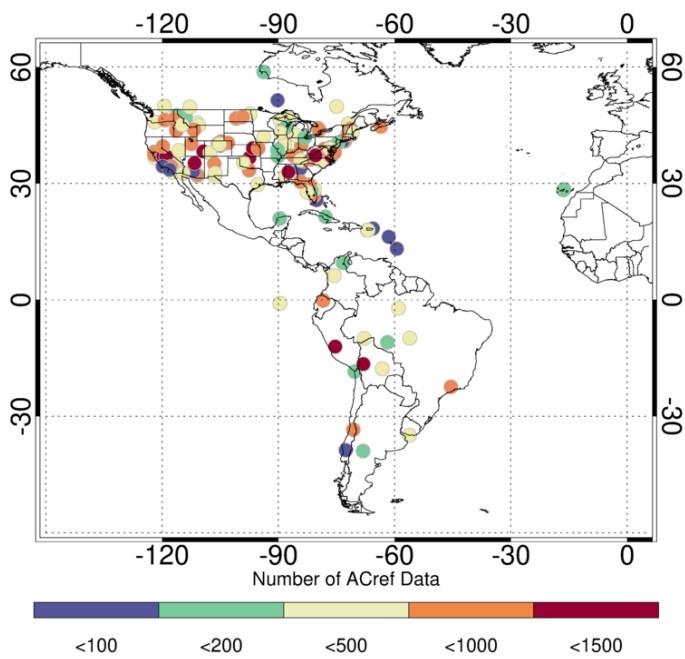

Number of ACref Data

| <100 | <200 | <500 | <1000 | <1500 |

**Figure 1 Map of AERONET which have available observation (AOD at 550 nm < 0.1, AE > 1.0) between July 2019 and June 2020, the time window for ABIE AC-ref calculation. The color represents number of valid observations which meet the criteria and applied to the AC_ref calculation.**

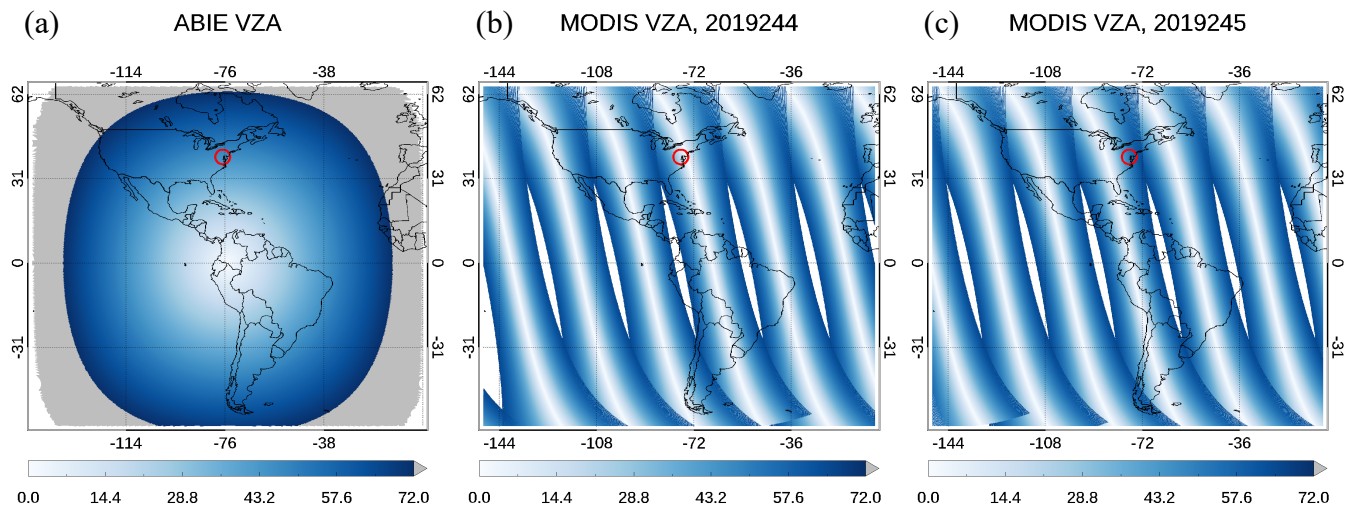


**Figure 2 (a) Full disk coverage of ABIE with VZA (< 72°) for each pixel and (b, c) MODIS VZAs for the same region on two consecutive days (Sep 01-02, 2019). The red circle indicates the location of GSFC AERONET site [-76.84°E,38.99°N].**

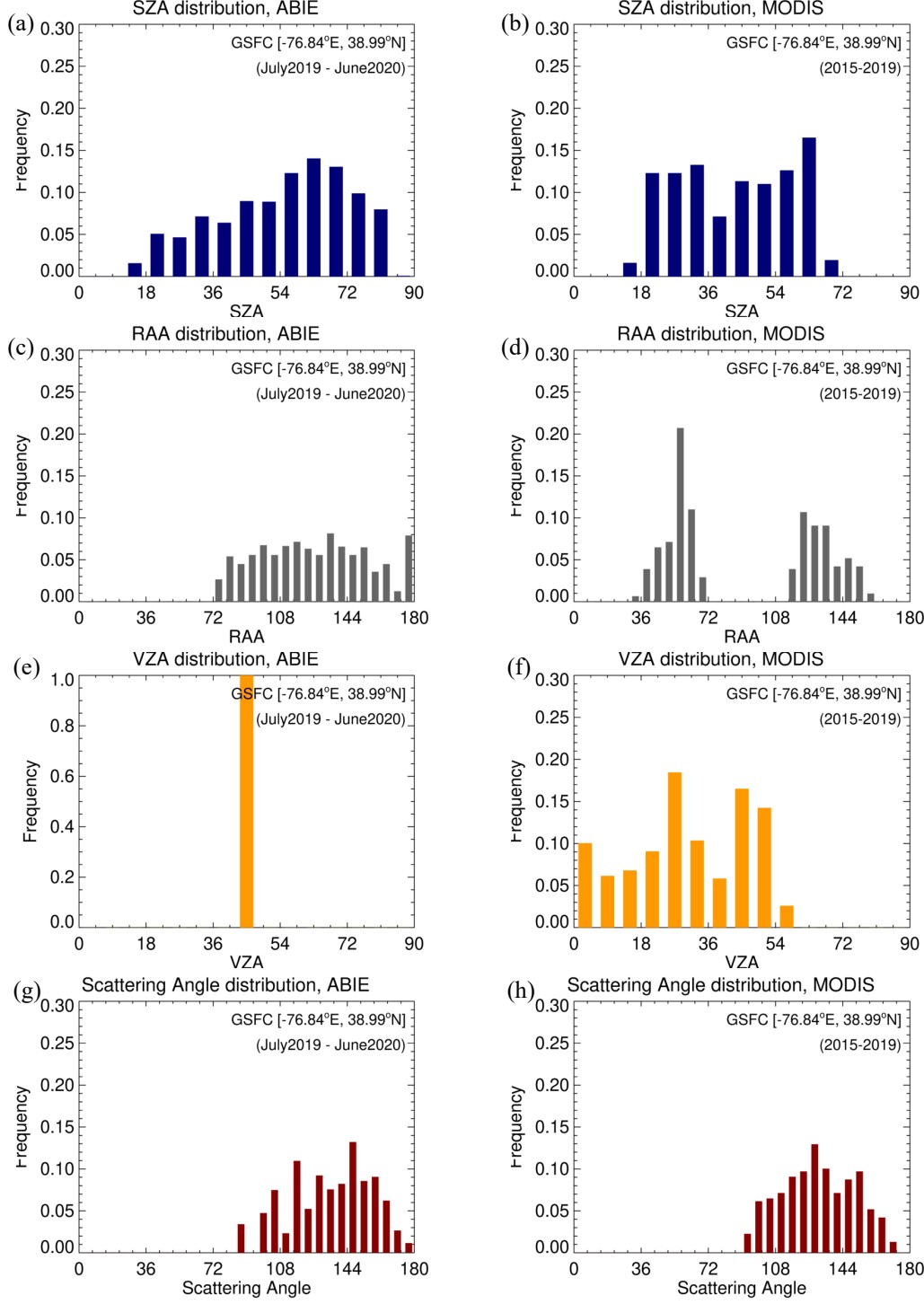

**Figure 3 Frequency distributions of sun-and-sensor geometries for (left) ABIE and (right) MODIS measurements at GSFC AERONET site: (a, b) SZA, (c, d) RAA, (e, f) VZA, and (g, h) scattering angle.**

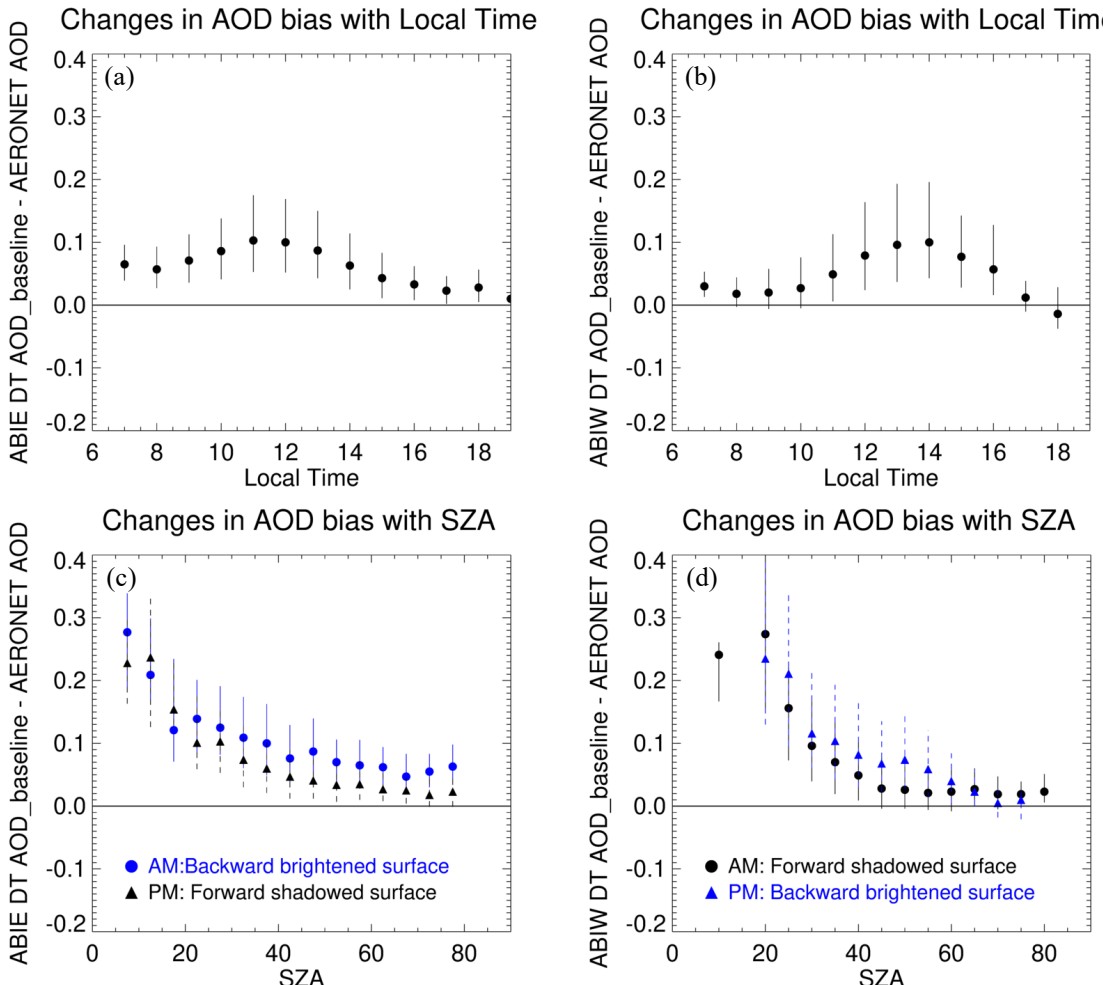

Figure 4 Bias of DT AOD compared to AERONET values. The DT AODs are retrieved from (a, c) GOES 16 ABI (ABIE) and (b, d) GEOS 17 ABI (ABIW) from August to September 2019. The biases are then calculated using data from all available AERONET site and aggregated for each (a, b) local solar time and (c, d) SZA. The symbol represents the median value and vertical bar displays the range between the 1st and the 3rd quartile of the AOD bias at each bin. In the bottom panels, the biases derived for morning and afternoon are differentiated by symbol [Circle: AM, Triangle: PM]. The blue represents backward scattering direction, while black indicates forward scattering direction.

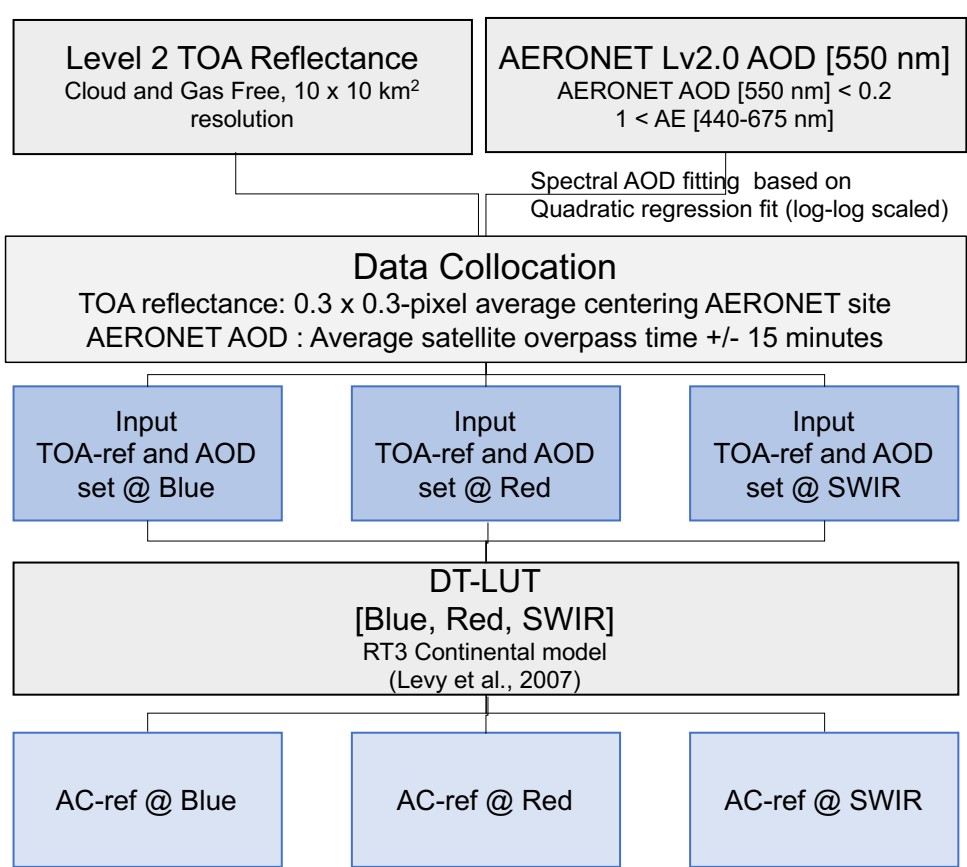

**Figure 5 Flowchart of calculating atmospherically-corrected reflectance (AC-ref). The DT-LUT was created using the RT3 code, assuming a black Lambertian surface.**


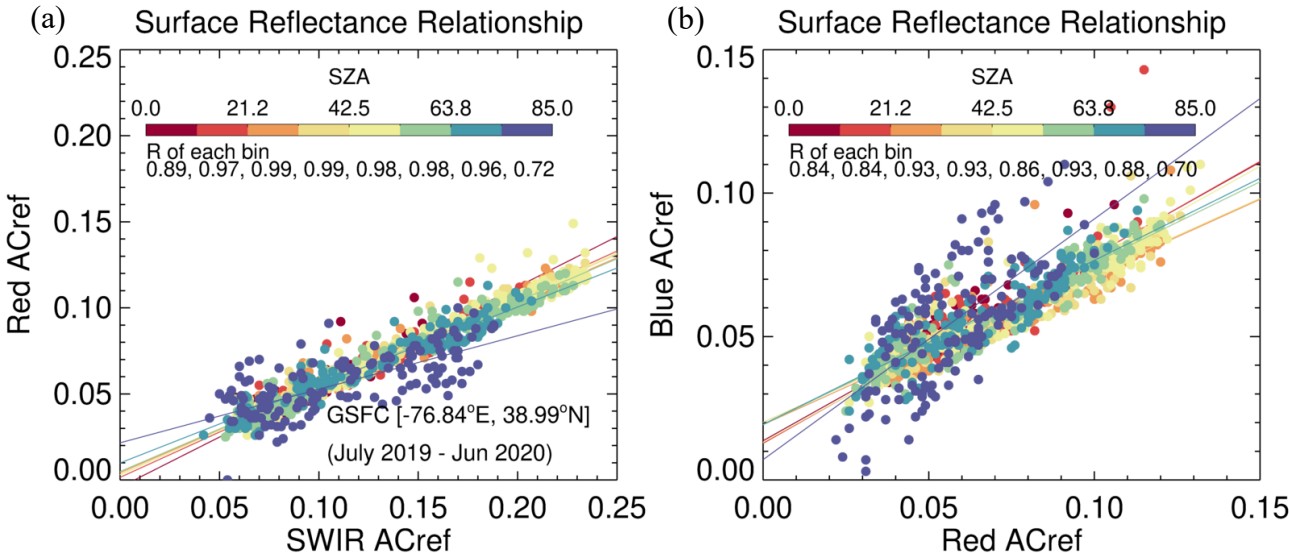

**Figure 6 Relationships between (a) Red and SWIR AC-ref and (b) Blue and Red AC-ref obtained from the GSFC AERONET site from July 2019 to June 2020. Color represents SZA and solid lines show linear regression for eight SZA groups. Each group has equal data points (144).**


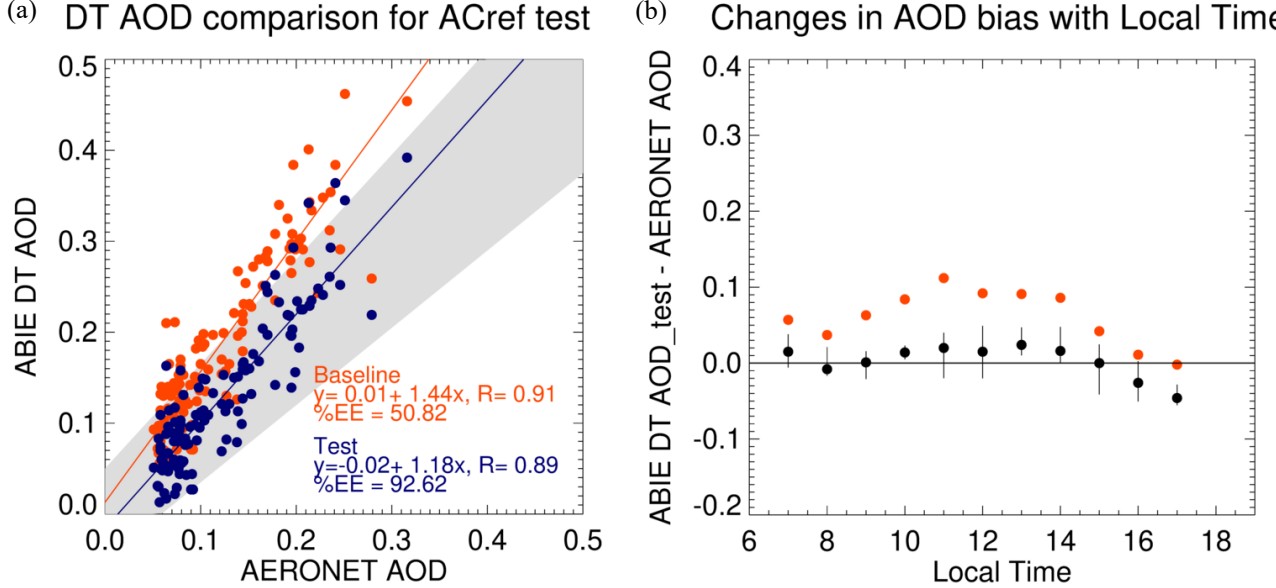

**Figure 7 Validation of DT AOD retrieved from ABIE at GSFC in September 2019; (a) Scatter plots between ABIE AOD and AERONET AOD and (b) diurnal changes in the bias between them. Orange and black indicate different assumption in surface reflectance parameterization (SRP); Orange represents the DT AOD retrieved assuming the baseline DT SRPs but blue represents the tested AOD adopting the SRPs obtained from the AC-ref of ABIE (Fig. 6). The symbol indicates the median of the bias, and the vertical bars represent the first and the third quartiles.**


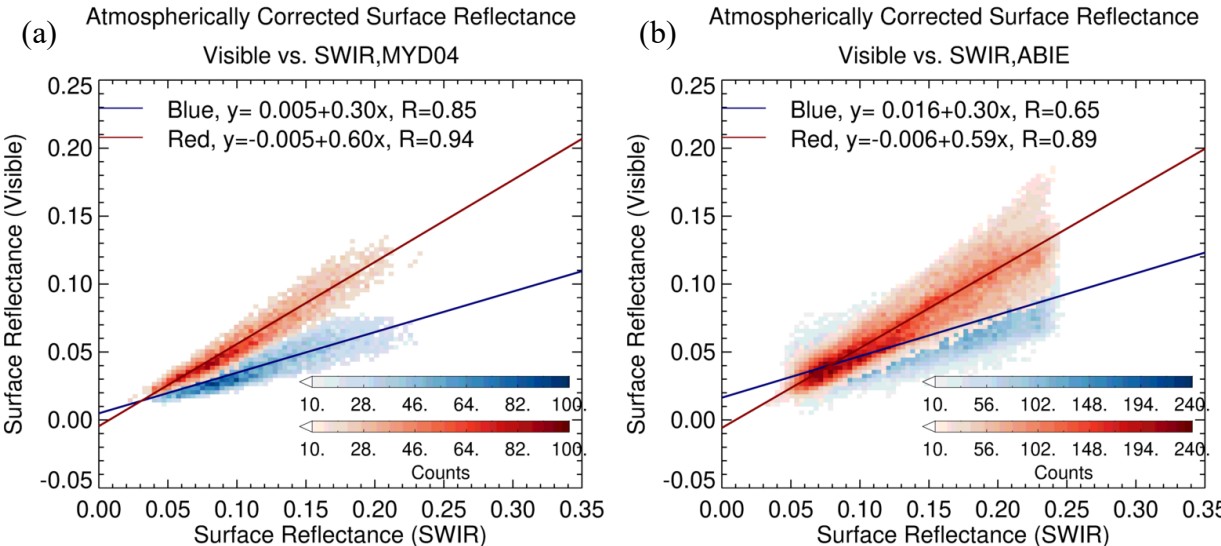

**Figure 8** AC-ref in the visible (0.47 and 0.66 μm for MODIS and 0.47 and 0.63 μm for ABIE) compared with SWIR AC-ref (2.12 μm for MODIS and 2.24 μm for ABIE). The (a) MODIS AC-ref consists of 5 years of observations from 2015 to 2019 over the ABIE Field of View while the (b) ABIE AC-ref is for one year from July 2019 to June 2020. Blue and red indicate Red-SWIR and Blue-SWIR relationship, respectively.


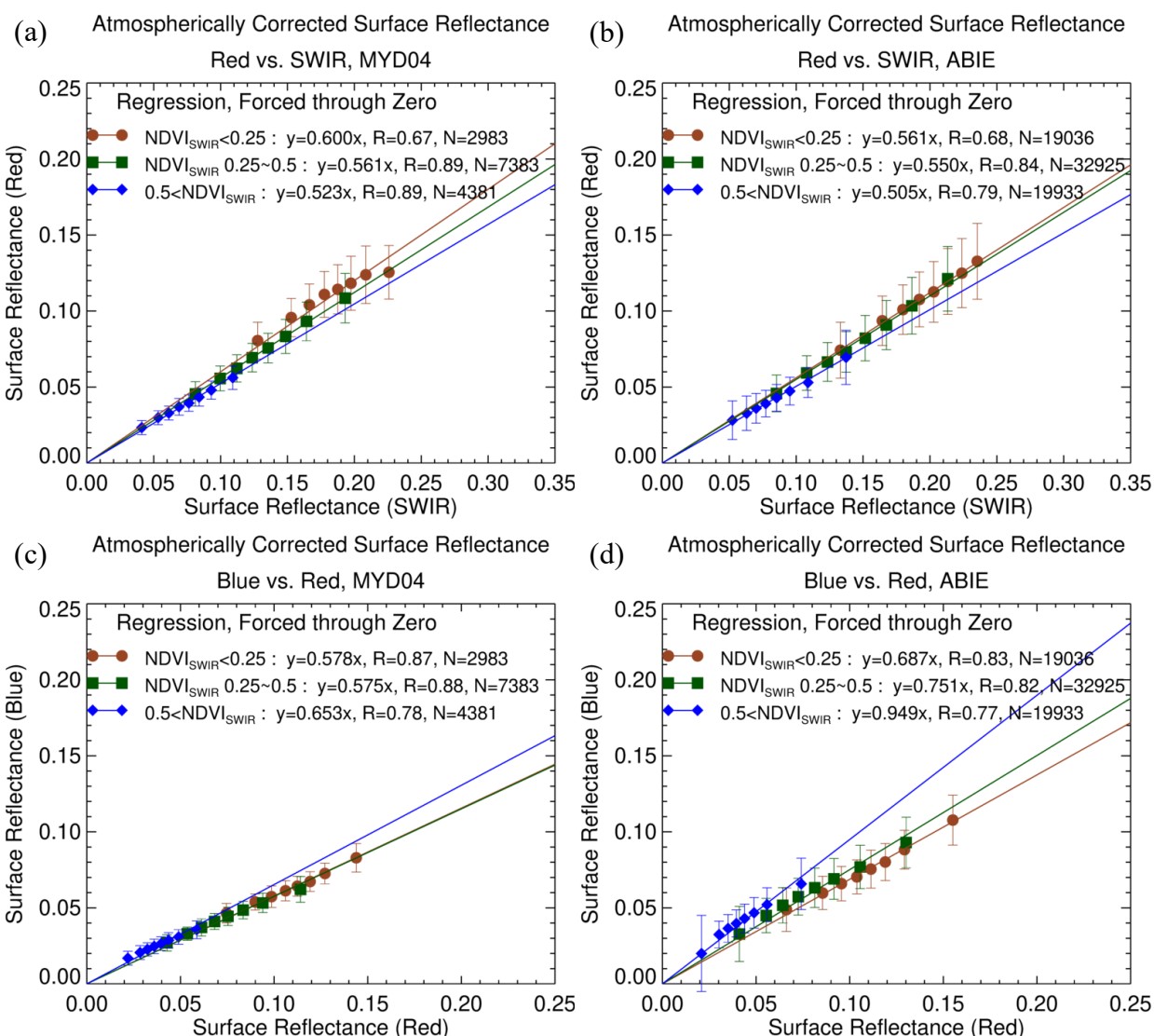

**Figure 9 The AC-ref relationships (a, b) between red and SWIR and (c, d) between blue and red channel as a function of NDVI$_{SWIR}$. Left and Right columns indicate the relationships for MODIS and ABIE, respectively. The regression equation is forced through zero. Red refers to low NDVI$_{SWIR}$, green to medium and blue to high values. Each AC-ref group for NDVI$_{SWIR}$ is divided equally into 8 bins and is displayed by mean (symbol) and standard deviation (vertical bar) of each bin.**

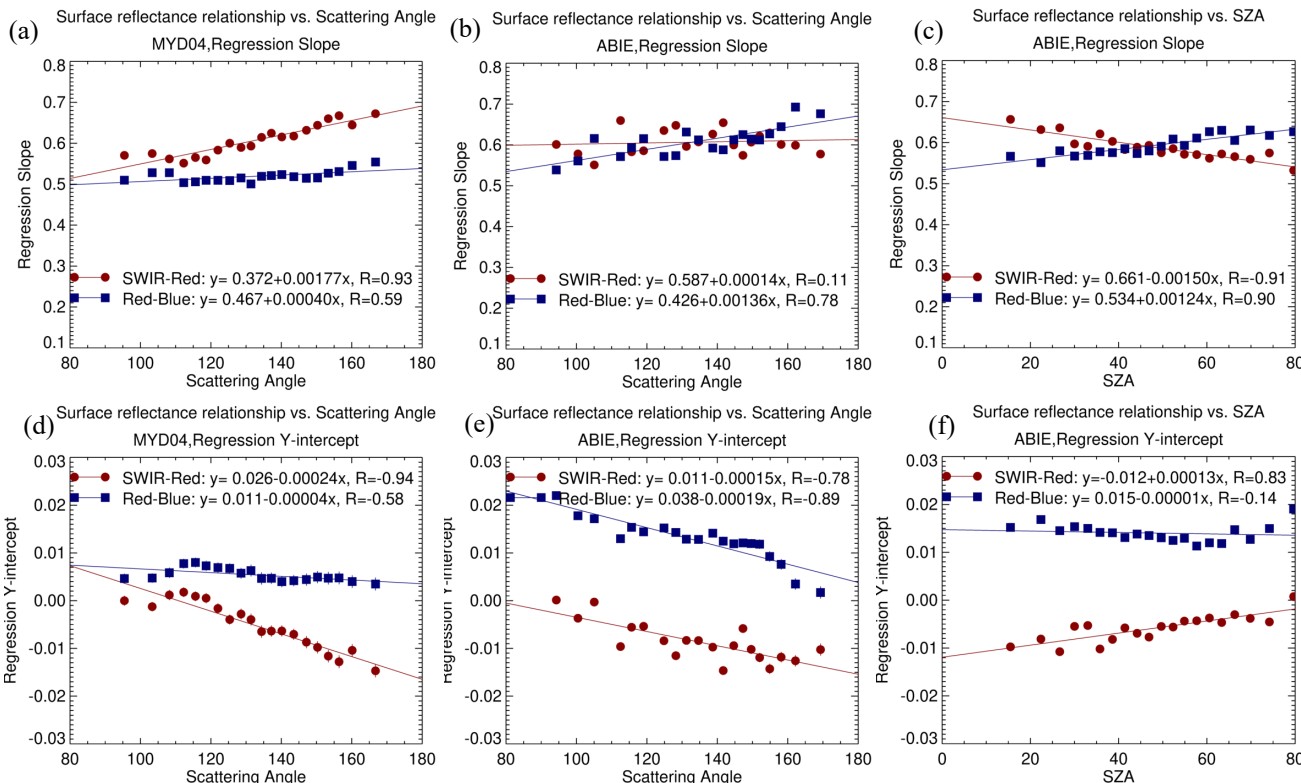


**Figure 10 Regression between visible and SWIR AC-ref of (left) MODIS and (middle and right) ABIE. Slope (a, b, c) and y-intercept (d, e, f) of regression are plotted as a function of (left and middle) scattering angle and (right) SZA. The data were sorted according to scattering angle/SZA and put into 20 groups of equal size (736 for MODIS and 3899 for ABIE). Blue square and red circle indicate red-SWIR and blue-red ratio, respectively.**


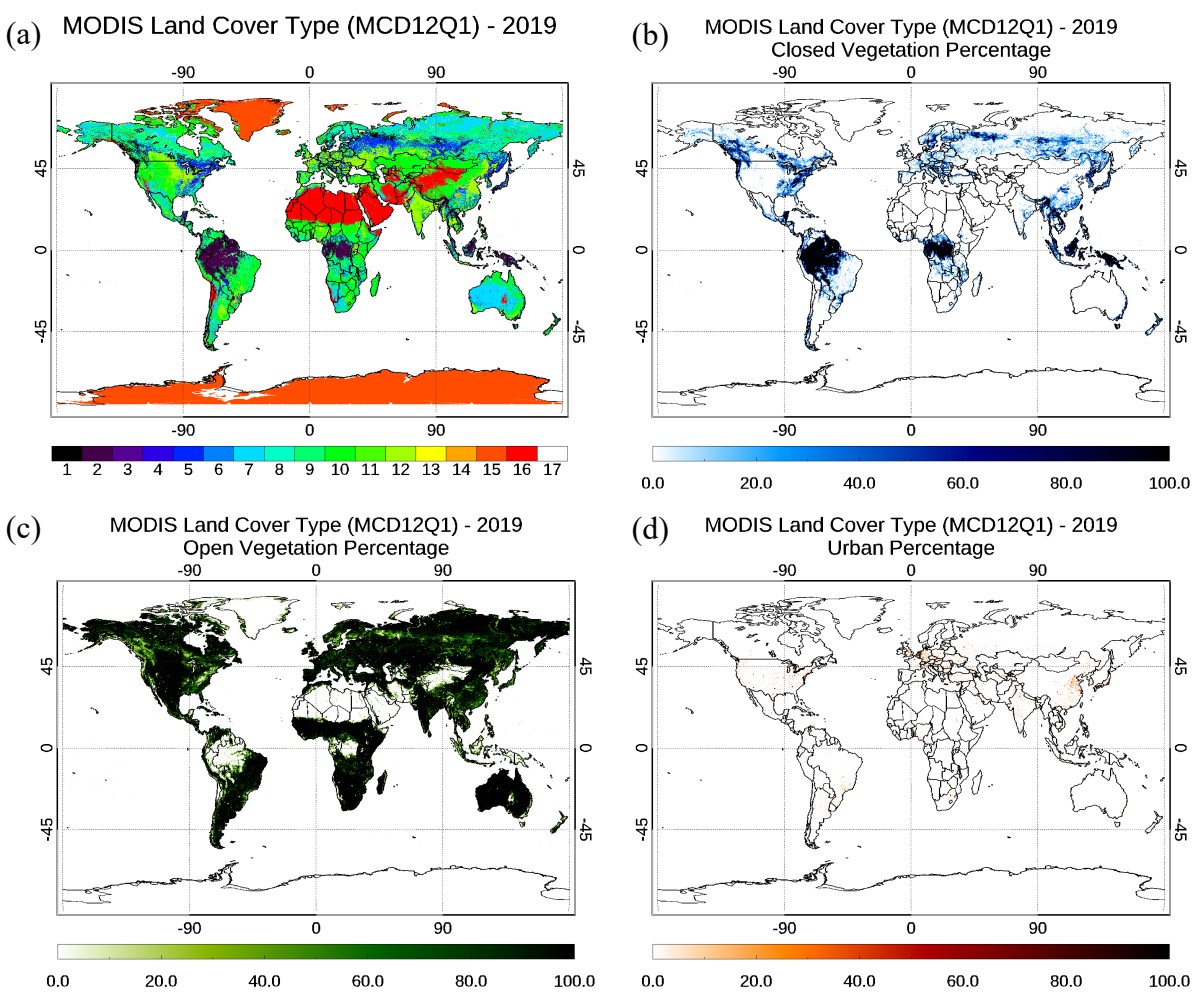

**Figure 11 A global map of (a) the most frequent International Geosphere Biosphere Program (IGBP) index within 0.1 x 0.1º grid box and the percentage of the sub-grid index classified into (b) Closed Vegetation, (c) Open Vegetation, and (d) Urban type. The IGBP index indicates 1. Evergreen Needleleaf Forests, 2. Evergreen Broadleaf Forests, 3. Deciduous Needleleaf Forests, 4. Deciduous Broadleaf Forests, 5. Mixed Forests ,6. Closed Shrublands, 7. Open Shrublands, 8. Woody Savannas, 9. Savannas, 10. Grasslands, 11. Permanent Wetlands, 12. Croplands, 13. Urban and Built-up Lands, 14. Cropland/Natural Vegetation Mosaics, 15. Permanent Snow and Ice, 16. Barren, 17. Water Bodies.**


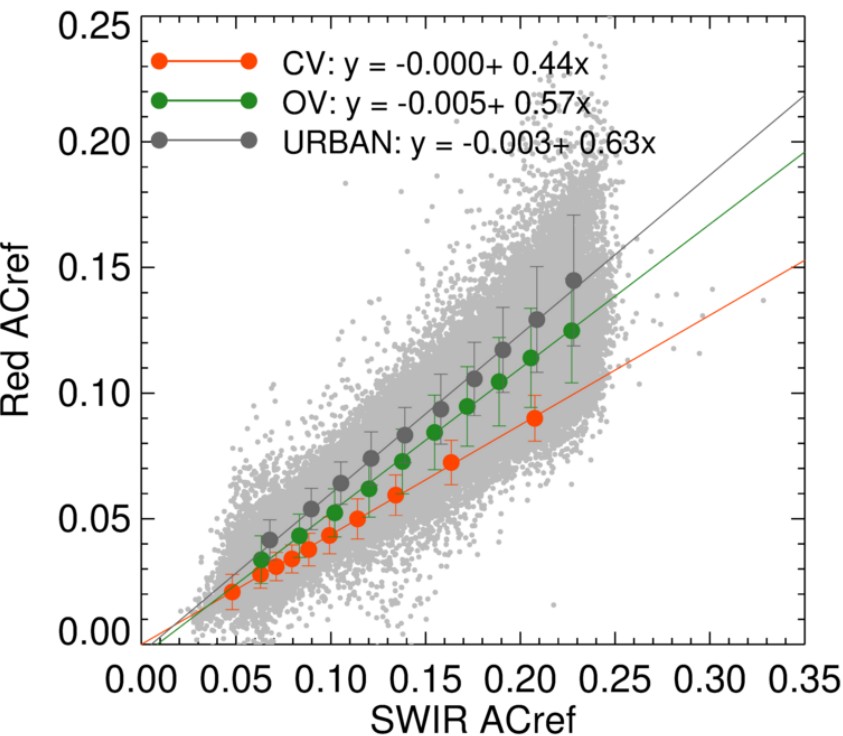

**Figure 12 Linear regressions (a) between Red and SWIR AC-ref classified into three land cover types: Closed Vegetation (CV, Red), Open Vegetation (OV, Green) and Urban (Black). Each AC-ref group is divided equally into 10 bins and is displayed by mean (symbol) and standard deviation (vertical bar) of each bin. The gray dot represents a population of AC-ref, which is not distinguished by land type.**


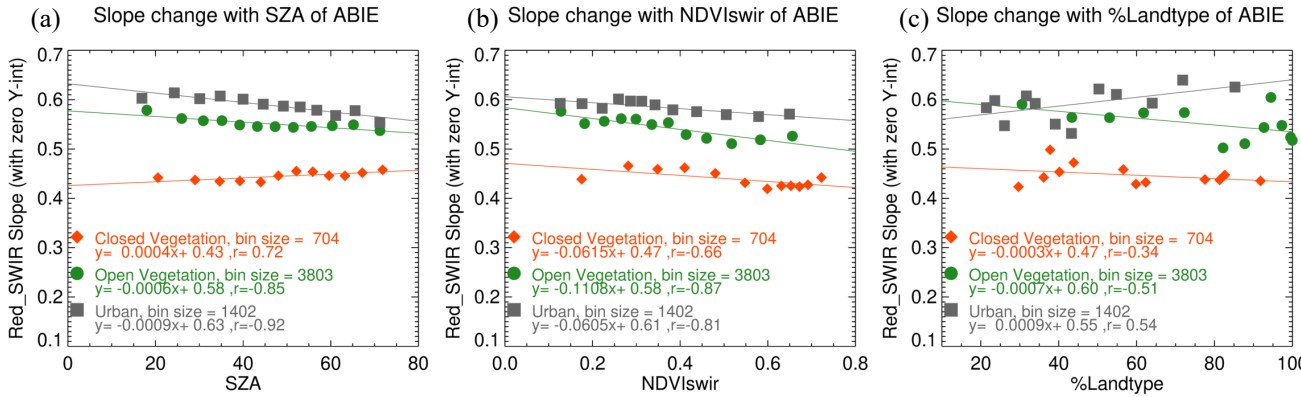


**Figure 13 The slope of regression between Red and SWIR AC-ref. The slopes are calculated for three land types, and shows changes depending on (a) SZA, (b) NDVI$_{SWIR}$, and (c) %land type. For each group of Closed Vegetation (CV, Red), Open Vegetation (OV, Green), and Urban (Gray), AC-ref are equally divided into 12 bins according to the x-parameter, respectively. Among the 193 AERONET sites used for the collocation, 40 sites corresponded to CV, 169 sites to OV, and 55 sites to urban.**


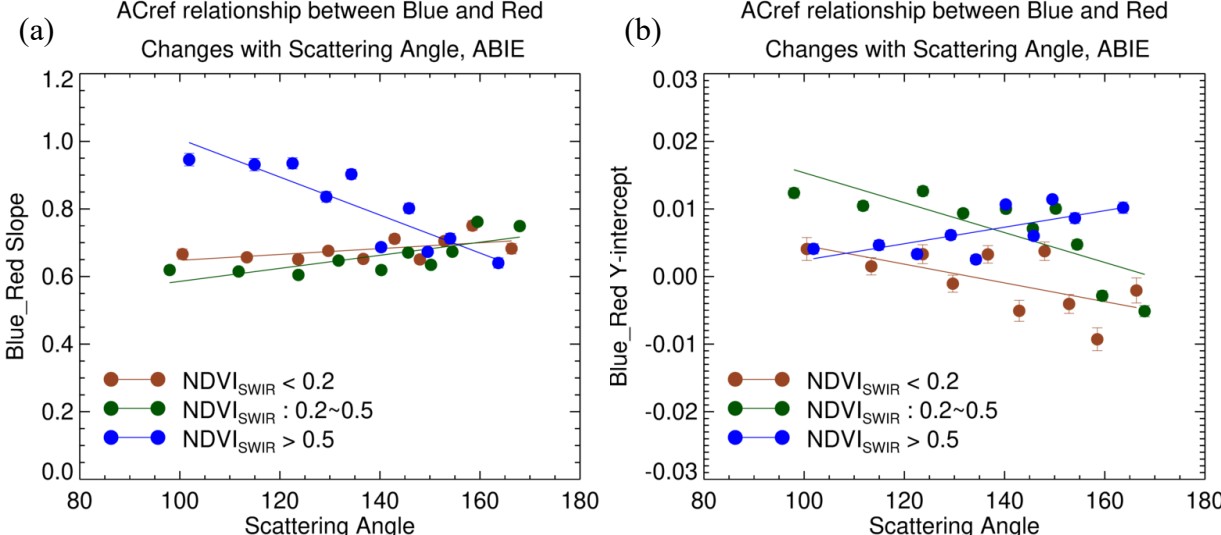

**Figure 14 The change of blue and red relationship with NDVI$_{SWIR}$ and scattering angle. Color indicates NDVI$_{SWIR}$ which is divided into three levels.**

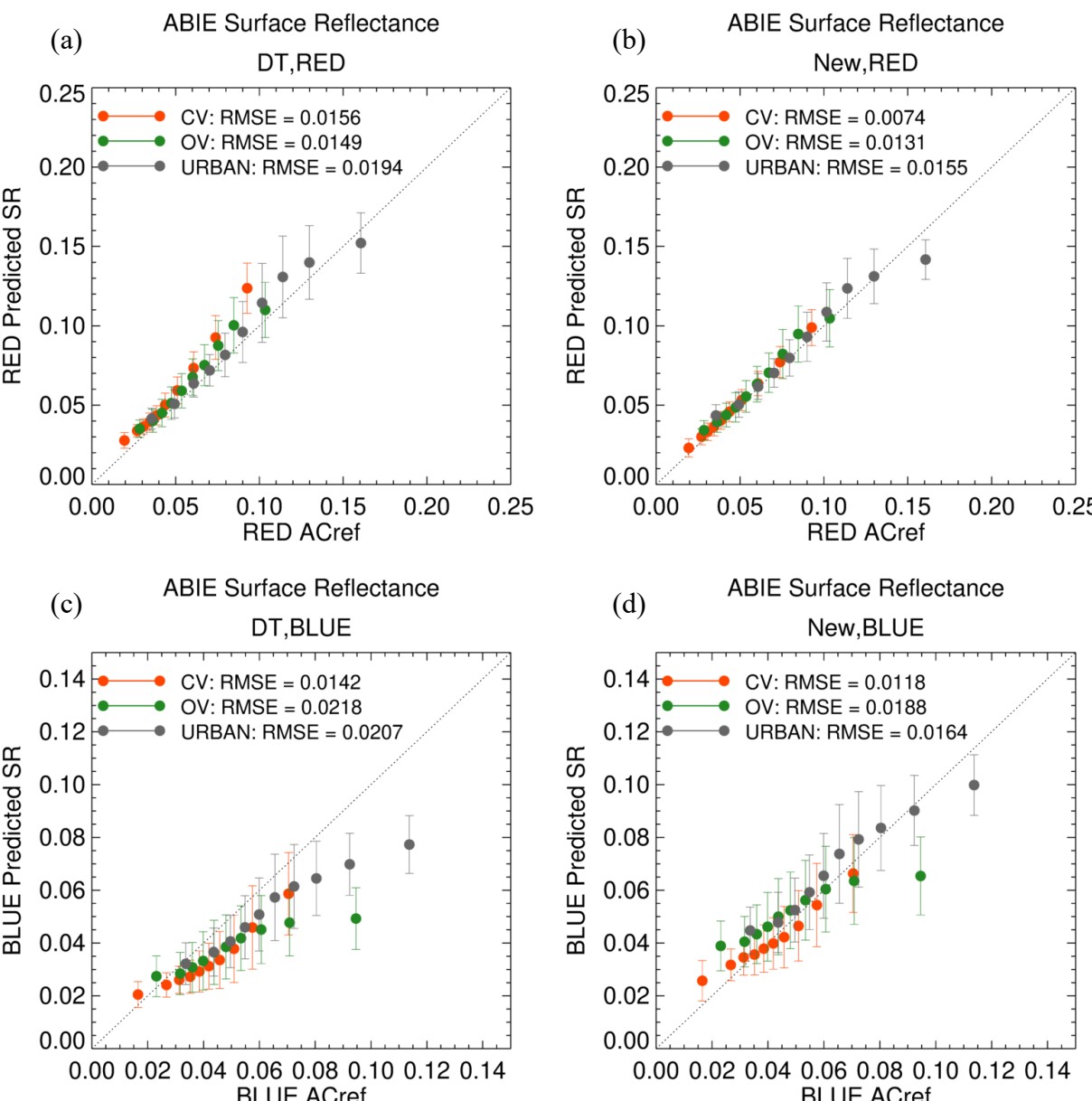

Figure 15 Comparisons between predicted surface reflectance and AC-ref for the (top) red and (bottom) blue channels. Left panels show the predicted surface reflectance achieved from the baseline DT SRP and the Right panels show the predicted surface reflectance using the new DT-GEO SRP. The predicted surface reflectance is sorted by AC-ref and equally divided into 10 bins. Symbol and vertical bar represent mean and standard deviation in each bin respectively, and color indicates land type: (red) CV, (green) OV, and (gray) Urban.

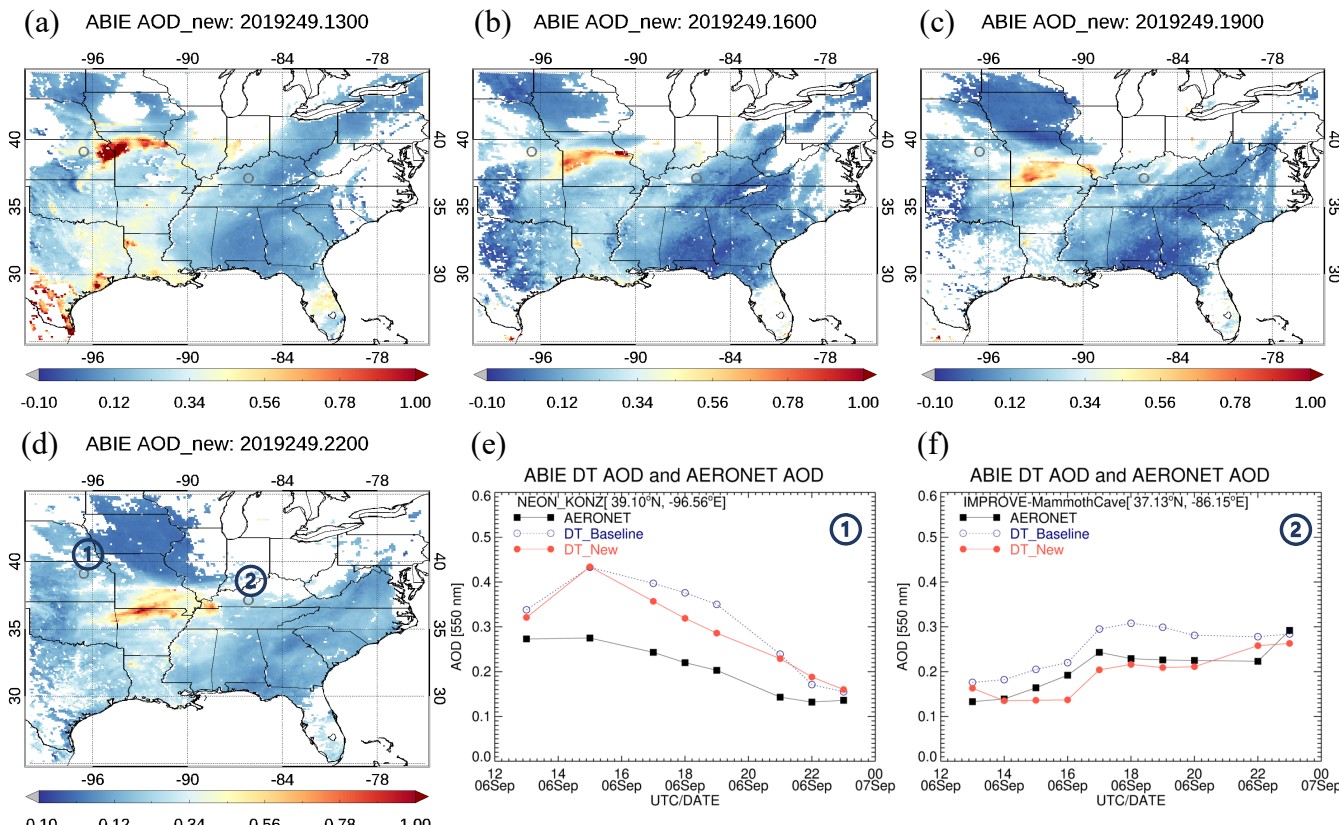

**Figure 16 Three-hour interval map (a-d) displaying the new DT AOD retrieved from ABIE on September 6, 2019. A comparison is made between the new DT AOD and the baseline DT AOD as well as the AERONET AOD at two locations: (e) NEON_KONZ [39.10°N, -96.56°E] and (f) IMPROVE_MammothCave [37.13°N, -86.15°E]. A Black square indicates AERONET AOD [550 nm], a blue open circle represents the baseline DT AOD [550 nm], and a red closed circle denotes the new DT AOD [550 nm]. The AERONET sites are marked on the map with gray circle.**

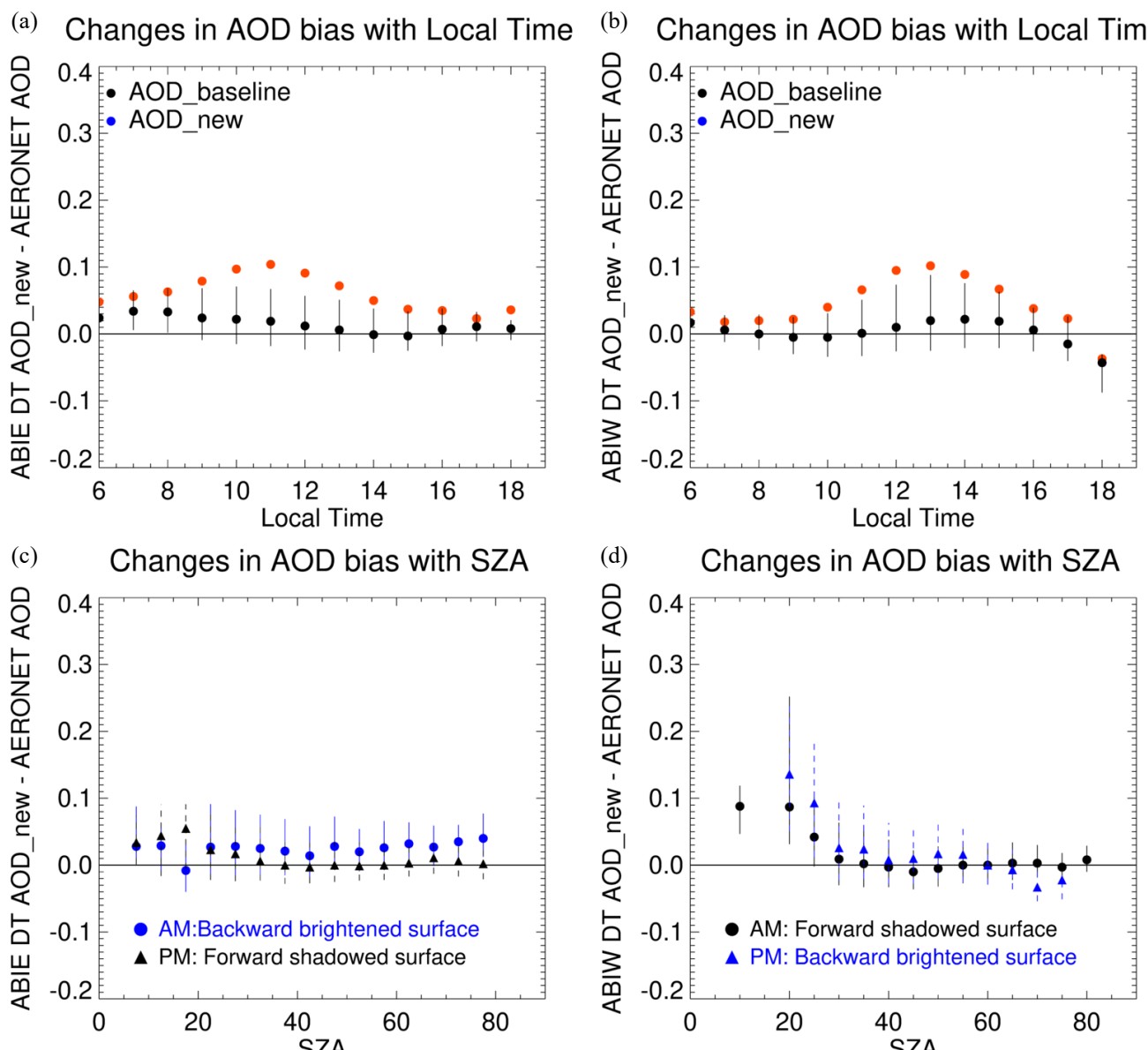

**Figure 17 Comparison of the bias in the DT baseline AOD and DT new AOD. The DT AODs are retrieved from (a, c) GOES 16 ABI (ABIE) and (b, d) GEOS 17 ABI (ABIW) from August to September 2019. The biases are then calculated using the collocated AERONET AOD from sites within the sensors' disk scan and aggregated by (top) local solar time and (bottom) SZA. In the upper panels, the biases in the new DT AOD (Black) are overlaid on the biases of (Red) baseline AOD. The symbol represents the median value and vertical bars display the range between the 1st and the 3rd quartile of the AOD bias at each bin. The bottom panel shows the bias range of the DT new AOD for each SZA bin (10° interval division). The bias derived for morning and afternoon and differentiated by symbol [Circle: AM, Triangle: PM]. The blue represents backward scattering direction, while black indicates forward scattering direction.**



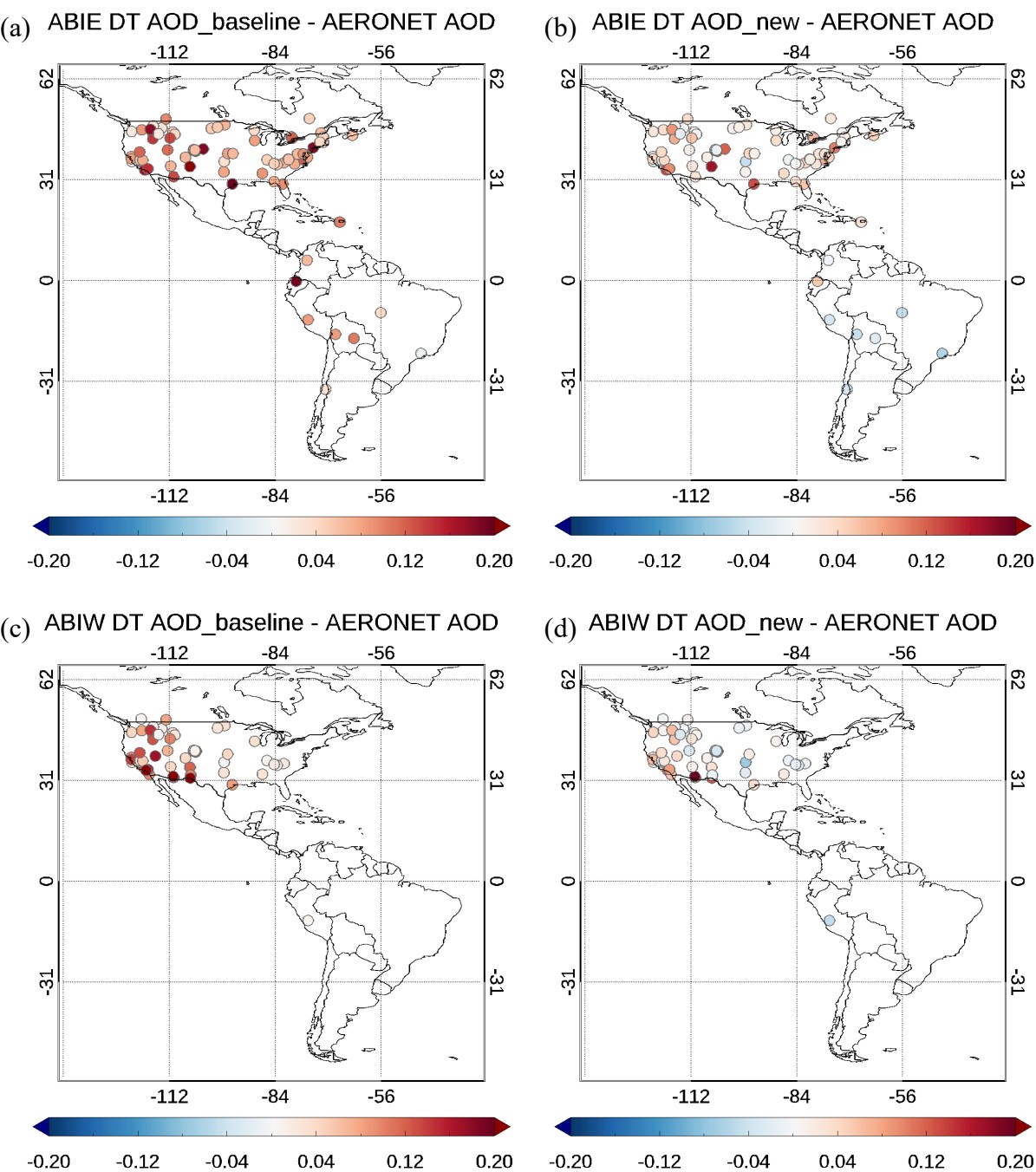

**Figure 18 Bias maps of (a, c) the baseline DT AOD and (b, d) the new DT AOD retrieved from (a, b) ABIE and (c, d) ABIW for August - September 2019. The bias represents the average of the absolute difference between DT AOD and AERONET AOD.**


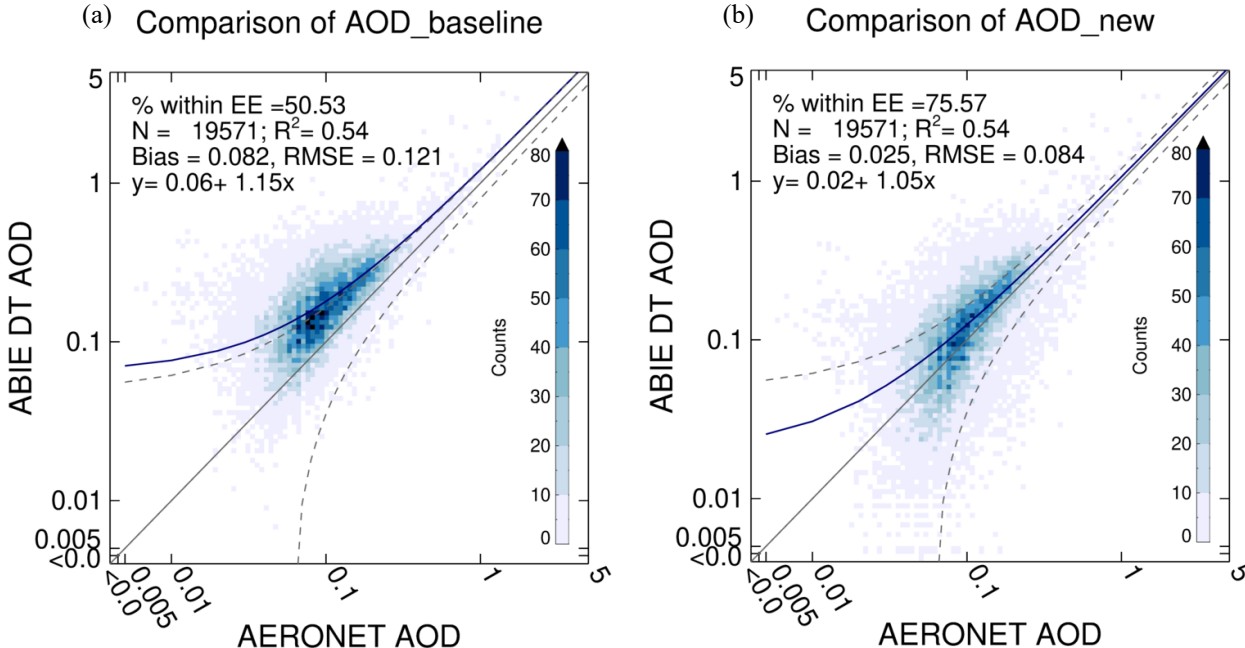

**Figure 19 Comparison of the AEORNET AOD and the DT AOD retrieved from ABIE from August to September 2019 across all collocated AERONET stations within the sensors' disk scan. The DT AOD are retrieved by applying (a) the baseline and (b) the new SRP, respectively. The color indicates number density of scatter plot.**


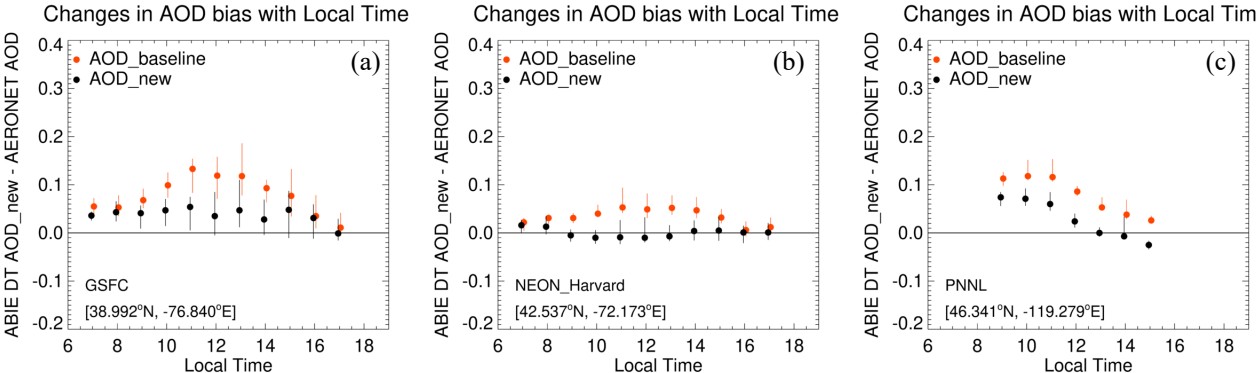

**Figure 20 Comparison of the bias in the DT AOD at GSFC, NEON_Harvard, and PNNL AERONET site. The bias is calculated from the DT baseline and new retrieval from August to September 2019. The range of bias for each time is expressed as a vertical bar connecting the first and third quartiles., and the median bias is shown by symbol. Color indicates the baseline (Red) and the new AOD (Black).**


**Table A1. Table A1 List of location used for the NDVI$_{GEO\_SWIR}$ and NDVI$_{LEO\_SWIR}$ comparison test.**

| Location | Latitude ($^O$N) | Longitude ($^O$E) | Dominant landtype (IGBP classification) | %landtype |
|---|---|---|---|---|
| Alta_Floresta | -9.87 | -56.10 | Savannas | 45.64 |
| Amazone_ATTO_Tower | -2.14 | -59.00 | Evergreen Broadleaf forests | 98.36 |
| GSFC | 38.99 | -76.84 | Urban and Buit-up | 44.09 |
| La_Paz | -16.54 | -68.07 | Grasslands | 74.46 |
| NEON_WREF | 45.82 | -121.95 | Evergreen Broadleaf forests | 93.33 |
| Rio_Branco | -9.96 | -67.87 | Savannas | 62.78 |


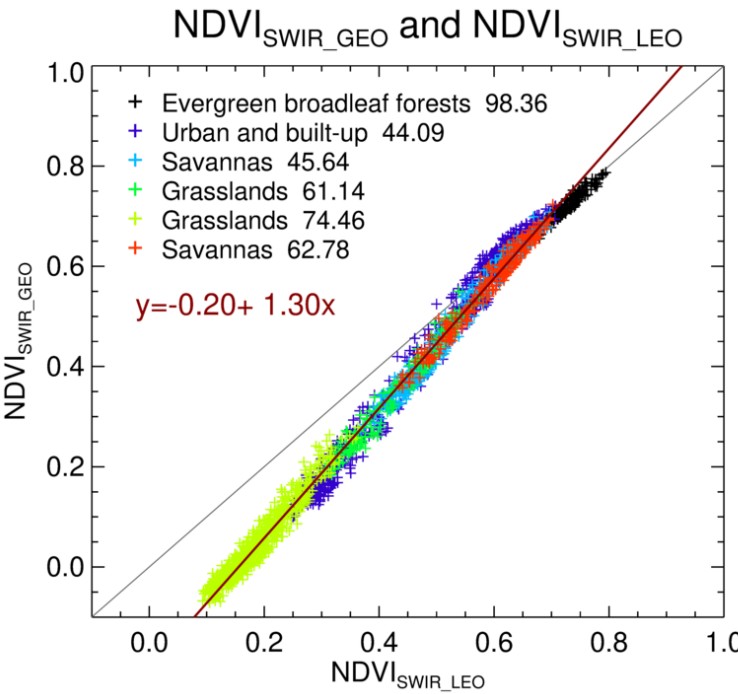

**Figure A1 Comparison of NDVI$_{GEO\_SWIR}$ and NDVI$_{LEO\_SWIR}$ derived from MODIS (Aqua) TOA reflectance at six locations from Jan 2017 to June 2020. Color indicates the location. The names and numbers shown in the legend indicate the dominant land types and their occupancy at each location.**

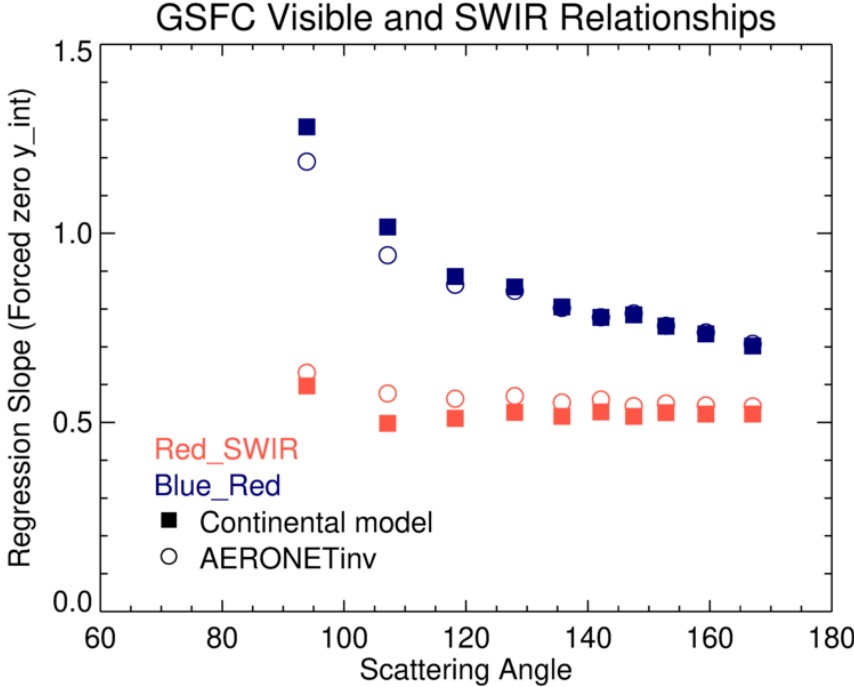


**Figure B1 The changes of visible vs. SWIR relationship with scattering angle. The red and blue represent Red vs. SWIR and Blue vs. Red relationship, respectively, and the symbol compared the aerosol model assumption in the AC-ref calculation. Square: LUT-based continental model assumption, Circle: 6sV-based inversion of AC-ref using the AERONET inversion products.**