# Peer review of "Parameterizing spectral surface reflectance relationships for the Dark Target aerosol algorithm applied to a geostationary imager"

_Atmospheric Measurement Techniques, 2023_

## Author Response (AR1)

Reviewer #1.

This study presents an adaptive improvement to apply the DT method to ABI/GEOS sensors, which follows the general scientific method and principles, and the results are reasonable and scientifically significant. The study specifically points out and discusses the differences in observation geometry between geostationary and polar-orbiting satellites when applying the DT method. It further subdivides the impacts caused by underlying surface types, which is of great significance. These contributions can also assist future satellite aerosol retrieval algorithms and match with the journal's objectives. However, the existing article structure lacks clarity, and paragraph coherence is weak, requiring improvements in the manuscript's organization. Additionally, there are some technical issues that need clarification.

We appreciate your thoughtful comments and suggestions. Our responses to each comment are provided below.

Specific Comments:

- In the end of Abstract, I recommend to add a sentence to describe the final validation results of the DT/ABI method, such as several evaluation indicators: R, bias, RMSE, and %EE.

Statistics of the validation including R, linear regression, and %EE will be added to the end of Abstract on lines 27-29 as follows.

*"The agreement between DT and AERONET AOD is established through regression slope of 1.06 and y-intercept of 0.01 with correlation coefficient is 0.74. By using the new SRP, the percentage of data falling within the expected error range (±0.05 + 15%) is notably risen from 54% to 78%."*

- It makes me slightly confused that the authors use quiet large text to describe existing DT/MODIS method in section 2.1 from Line 91 to 191, including strategies, equations, and product details. Though the research is based on the DT method, too many reproductions of existing literatures will make the manuscript look like a student technical report rather than a scientific study.

Section 2.1 has been revised to describe DT method with a focus on surface reflectance. Other details have been replaced by references.

- Line 220: How to screen cloud from red band? Is there any product or existing method? Please clarify or add references, since cloud will greatly impact on aerosol retrieval.

Previous studies describe DT cloud and ice/snow mask for MODIS and GEO sensor has been added in the manuscript (on lines 215-217) as follow.

*"Cloud and ice/snow masking are modified to account for lack of 1.24 µm and some of the TIR bands. Details about the pixel masking for MODIS is described in DT ATBD*

*(https://darktarget.gsfc.nasa.gov/atbd-land-algorithm) and Shi et al. (2021), and the modification for GEO is described in Gupta et al. (2019)."*

- Line 225. Using 0.86 to replace 1.24 is a good choose. I prefer to a similar indicator named 'AFRI' (10.1016/s0034-4257(01)00190-0), and it shows a good consistency (slope close to 1) with MODIS NDVISWIR and shows relatively good atmospheric resistance (10.1109/TGRS.2020.3021021). The discussion of the NDVI index selecting should be extended here because it is one of the key steps of the DT method, and this problem also usually appears when applying the DT method to other sensors.

Thanks for this comment. $NDVI_{GEO\_SWIR}$ used is based on the $NDVI_{MIR}$ defined in Karnieli et al. (2001) and Miura et al., 1998. To clarify the background of $NDVI_{GEO\_SWIR}$, we added the references [A. Karnieli et al (2001) ; Jin et al., 2021] on lines 224-225.

We also conducted a study to check a consistency between $NDVI_{LEO\_SWIR}$ and $NDVI_{GEO\_SWIR}$. The study is based on the MODIS TOA reflectance, which has both 0.86 μm and 1.24 μm channel. We derived the indices for 3.5 years (01.2017-06.2020) at six selected AERONET sites (Table A1) covering different land cover types.

The Fig. A1 shows $NDVI_{LEO\_SWIR}$ and $NDVI_{GEO\_SWIR}$ has a good consistency with linear regression. $NDVI_{GEO\_SWIR}$ matches $NDVI_{LEO\_SWIR}$ in dense vegetation where both $NDVI_{SWIR}$s are high but falls to lower values at the low end in scenes with less vegetation coverage. Accordingly, in the SRP, the $NDVI_{GEO\_SWIR}$ increases the number of cases assigned into the low $NDVI_{SWIR}$ category ($NDVI_{SWIR}$ < 0.25) from 25.5% to 41.5%. This means that without modifying the threshold on $NDVI_{GEO\_SWIR}$ the SRP will be encountering pixels never used before by the DT algorithm. The red surface reflectance will be parameterized differently in areas with less vegetation, which may lead to differences between DT GEO and DT LEO retrievals. It requires a new NDVI threshold or SRP improvement for the DT GEO retrieval.

The NDVI comparison has been added into Appendix A. and discussed in lines 225-232.

*Table A1 List of location used for the $NDVI_{GEO\_SWIR}$ and $NDVI_{LEO\_SWIR}$ comparison test.*

| Location | Latitude (°N) | Longitude (°E) | Dominant landtype (IGBP classification) | %landtype |
|---|---|---|---|---|
| Alta_Floresta | -9.87 | -56.10 | Savannas | 45.64 |
| Amazone_ATTO_Tower | -2.14 | -59.00 | Evergreen Broadleaf forests | 98.36 |
| GSFC | 38.99 | -76.84 | Urban and Buit-up | 44.09 |
| La_Paz | -16.54 | -68.07 | Grasslands | 74.46 |
| NEON_WREF | 45.82 | -121.95 | Evergreen Broadleaf forests | 93.33 |
| Rio_Branco | -9.96 | -67.87 | Savannas | 62.78 |

[Figure]

*Figure A1 Comparison of NDVI$_{GEO\_SWIR}$ and NDVI$_{LEO\_SWIR}$ derived from MODIS (Aqua) TOA reflectance at six locations from Jan 2017 to June 2020. Color indicates the location. The names and numbers shown in the legend indicate the dominant land types and their occupancy at each location.*

- Line 250: '…. is greater than' what?

The sentence has been revised on line 251. It should be "Ångström exponent (AE) (0.44 – 0.675 µm) is greater than 1.0".

- Line 255: why the spatiotemporal criteria are different between AC and validation? The motivation is unclear for some operational details.

Collocation for AC-ref calculation and verification use different criteria for spatial matching. We used a wide spatial range for the TOA-ref collocation.

 We've added a description on lines 256-265 as follows:

*"Spatiotemporal criteria for the satellite-AERONET co-locations are as follows. For AC-ref calculation, AERONET observations within ±15 minutes of satellite overpass (MODIS or ABIE) are collocated with satellite-derived TOA reflectance within ±0.3° rectangular grid centered over an AERONET site. Here, a relatively large spatial range was established to capture TOA reflectance influenced by diverse land cover types and to mitigate potential cloud contamination. The AC-ref dataset consists of the spatial mean of TOA reflectance, spectral AOD from AERONET, land cover types present with the range, and observation geometries. Figure 1 displays number of data used for the AC-ref calculation at each AERONET site. For the purpose of AOD validation, a temporal criteria of ±15 minutes for AERONET AOD and a spatial criterion of ±0.2° rectangular grid for DT AOD are applied. The spatiotemporal collocation window follows standard DT validation practice used in the MEaSUREs program. "*

- Section 3.1: From figure 2 (g-h), the distribution of scattering angle is quite similar between ABIE and MODIS, though the solar and sensor geometries are different. In the DT method, the estimation of SPR is based on the scattering angle, not the SZA, RRA, or VZA. Therefore, the potential impacts of these differences in observational geometry in retrieval are still not clear. In addition, (Line 290) I prefer to say that the MODIS SZA is relatively evenly distributed between 20 and 70; And there is a consensus that a high SZA is not conducive to the retrieval.

Thanks for the comment. Line 299 has been modified based on reviewer suggestions. We agreed that the difficulties posed by geometric differences were unclear from Fig. 3. Thus, we have added a discussion of the difficulties in AOD retrieval that may arise due to GEO observation geometry in as follows (lines 305-315).

*"Based on a comparison of observation geometries, it appears that LEO observation using the DT technique is more favourable to retrieve AOD than GEO observations. First, high SZA observations introduce greater uncertainty in AOD retrieval. Increased path length at high SZA makes it difficult to separate the aerosol contribution from other atmospheric components. The high SZA also can lead to uncertainty in AC-ref calculations and make the SRP analysis more difficult. Second, GEO observation are more likely to observe the 'vegetative hot-spot'. This is when the solar direction coincides with the observation direction ($\Theta \geq 175°$), resulting in a large increase surface reflectance at each wavelength. (Li et al., 2021) shows that ABI observe hot-spot frequently. The brighter surface overwhelms the aerosol contribution to the TOA reflectance and increases the uncertainty in AOD retrieval. Figure 3(g) also shows that ~4% of ABIE observations are performed at high scattering angles (>168°), whereas ~1% of MODIS observation reach that extreme at GSFC. The broad range of geometries lead to a difficulty in GEO AOD retrieval (Ceamanos et al., 2023) because the sensitivity to aerosols varies significantly during the day (Luffarelli and Govaerts, 2019; Ceamanos et al., 2019)."*

- Line 305-314: Yes, the discussion here is meaningful. This means we need to pay more attention to the details of the observational geometry than just the scattering angle. In addition, considering the particularity of the DT method, this probably also be only related to the instrument itself. For example, the AOD also has nonnegligible difference between MODIS/Terra and MODIS/Aqua. And in your study, the ABIE and ABIW also show difference in distributions of AOD bias as Figure 3. When applying the DT method to the AHI/Himawari (10.5194/amt-12-6557-2019), the dependence of AOD bias with time is not emphasized. So, the differences between different hardware are also large, and comparing with the time dependence, the overestimation of AOD (positive bias) may be more noticeable. In this case, it is necessary to re-formulate the SRP.

Thank you for bringing up this concern. Following the reviewer's feedback, we have revised section 3.2. Sensor calibration status can influence retrieval performance. However, if sensor specifications are the same, we do not anticipate significant biases due to hardware differences compared to the uncertainty in surface reflectance. To enhance our understanding of the difference in the AOD bias between ABIE and ABIW, we revised Fig. 4 to include the bias dependency on SZA (description on lines 320-328).

Figure 4(c,d) illustrates that the DT bias overall increases as SZA decreases, but with differences depending on the scattering direction between the Sun and the satellite. ABIE AOD exhibits a higher bias in the morning (Fig. 3c), as shown by the time dependence showcasing a distribution skewed towards the morning (Fig. 4a). In contrast, ABIW AOD shows a greater bias in the afternoon then the morning (Fig. 4d), aligning with the afternoon-skewed distribution (Fig. 4b). This time dependence is consistent in that the bias is relatively large when the direction in which the satellite faces the Earth's surface matches the direction in which sunlight arrives. In other words, the AOD bias is also influenced by the RAA, showing a reduced bias during low RAA conditions when the surface appears relatively dark due to shadows cast by the canopy. This means we need to pay more attention to the details of the observational geometry than just the scattering angle.

[Figure]

*Figure 4 Bias of DT AOD compared to AERONET values. The DT AODs are retrieved from (a, c) GOES 16 ABI (ABIE) and (b, d) GEOS 17 ABI (ABIW) from August to September 2019. The biases are then calculated using data from all available AERONET site and aggregated for each (a, b) local solar time and (c, d) SZA. The symbol represents the median value and vertical bar displays the range between the 1st and the 3rd quartile of the AOD bias at each bin. In the bottom panels, the biases derived for morning and afternoon are differentiated by symbol [Circle: AM, Triangle: PM]. The blue represents backward scattering direction, while black indicates forward scattering direction.*

- Line 345: Using extrapolation should be cautious and the data availability needs to be checked (such as excluding negative values). In addition, from a statistical point of view, the use of MODIS AOD products is also reasonable, which can greatly expand the AC sample number.

We revisited the data availability and revised the phrase (lines 374-376) to show the percentage of cases which are available to use $AOD_{1640}$ for a second order fitting. Among the aggregated cases meeting the collocation criteria [$0<AOD_{550}<0.2$ and $AE_{440\_675} >1$], 84% includes $AOD_{1640}$.

- Line 355: This is partially due to relatively large uncertainties of radiative transfer code when SZA is large.

That is true. In section 3.1, we added the following discussion (lines 305-315) on the uncertainty in AOD retrieval or AC-ref calculations that may arise when SZA is large.

*"Based on a comparison of observation geometries, it appears that LEO observation using the DT technique is more favourable to retrieve AOD than GEO observations. First, high SZA observations introduce greater uncertainty in AOD retrieval. Increased path length at high SZA makes it difficult to separate the aerosol contribution from other atmospheric components. The high SZA also can lead to uncertainty in AC-ref calculations and make the SRP analysis more difficult. Second, GEO observation are more likely to observe the 'vegetative hot-spot'. This is when the solar direction coincides with the observation direction ($\Theta \geq 175°$), resulting in a large increase surface reflectance at each wavelength. (Li et al., 2021) shows that ABI observe hot-spot frequently. The brighter surface overwhelms the aerosol contribution to the TOA reflectance and increases the uncertainty in AOD retrieval. Figure 3(g) also shows that ~4% of ABIE observations are performed at high scattering angles (>168°), whereas ~1% of MODIS observation reach that extreme at GSFC. The broad range of geometries lead to a difficulty in GEO AOD retrieval (Ceamanos et al., 2023) because the sensitivity to aerosols varies significantly during the day (Luffarelli and Govaerts, 2019; Ceamanos et al., 2019)."*

- Line360: Here is using the new SRP from AC-ref instead of the original MODIS SRP? Yes, AOD bias (overestimation) is suppressed but it still shows the time dependence.

The new SRP mentioned here is a preliminary result derived from a pilot study shown in Fig. 6. Because the VIS-SWIR correlation is low when SZA is high (>~75°), the newly derived SRP may not be efficient in improving AOD at high SZA. Additionally, we would like to mention that the remaining time dependence may arise due to other uncertainties. Luffarelli et al. (2019) and Ceamanos et al. (2023) show that, where the atmospheric path is short (SZA ~0°), the TOA reflectance becomes less sensitive to AOD, resulting in significant uncertainties in AOD retrievals around noon.

- Figure 9: I find the points in ABIE are more discrete, especially between SWIR and Red bands. This is possibly the impact from large SZA.

Thank you for bringing this to our attention. The discrete distribution of bins is influenced by the scattering angle distribution. For reference, here we showed a comparison of the frequency distribution of scattering angles obtained from the entire collocation dataset for both ABIE (left) and MODIS (right). In contrast to MODIS, which exhibits a uniform distribution between 100° and 160°, ABI angles are primarily concentrated in the range of 130° to 160°. As a result, the points analyzed in ABIE are relatively dense within the 130°-to-160° range, but sparse elsewhere.

[Figure]

*Frequency distribution of Scattering angle calculated from the AC-ref dataset for each ABIE and MODIS. The ABIE dataset covers the one year period from July 2019 to June 2020, and the MODIS dataset is for 5 year between 2015 to 2019.*

- I wonder whether the relationship between Blue, Red and SWIR bands changes with SZA (similar to the representation in Figure 8-9). This is meaningful to study the time dependence of AOD bias.

Thanks for this comment. We have added the changes with SZA as follows (Fig. 10, lines 442-444);

*"While the Red vs. SWIR regression shows weak dependence on the scattering angle, the regression changes linearly with SZA in Fig. 10(c) and Fig. 10(f). As SZA increases, the slope increases, and the interception decreases. The linear dependence is also evident in the blue-red relationship, although the points are slightly deviating from the linearity when SZA is high (>70°)."*

[Figure]

*Figure 10 Regression between visible and SWIR AC-ref of (left) MODIS and (middle and right) ABIE. Slope (a, b, c) and y-intercept (d, e, f) of regression are plotted as a function of (left and middle) scattering angle and (right) SZA. The data were sorted according to scattering angle/SZA and put into 20 groups of equal size (736 for MODIS and 3899 for ABIE). Blue square and red circle indicate red-SWIR and blue-red ratio, respectively.*

- Line 438: The title should be bold.

-Thanks, the title has been revised.

- Figure 12: This figure is interesting and shows the difference of SRP dependence between OV and CV. But what worries me is that the divergence between the data points is so large that it seems to outpace the trend itself. In addition, the SRP dependence does not show a clear change with percent of OV or CV. This is possibly owing to the features of OV and CV, which are not significant enough (IGBP is a climatological classification product). By contrast, the urban percent is an opposite.

We agree with the reviewer's comment. The Fig. 13 was modified to confirm the linearity of the regression change more clearly. Lines 515-526 of the manuscript has been modified as well.

*"In Fig. 13 we look for dependencies of the regression by plotting the slope against SZA, NDVI$_{SWIR}$ and the different land cover type percentages. In this figure, we simplified the regression to force the y-intercept to 0 and display the derived slope to make it easier to see the linearity of the regression change with the parameters. For vegetative surface, slopes from the red-SWIR regression range between 0.4 to 0.6, and slope variability is well captured by parameterization*

*using NDVI$_{SWIR}$. The slope of the CV group shows a weak dependence on the parameters, with a slight decrease observed with increasing NDVI$_{SWIR}$ (Fig. 13b, red) and %CV (Fig. 13c, red). When the surface becomes even darker, where NDVI$_{SWIR}$ is greater than ~0.6, the regression tends to lose its sensitivity to the NDVI$_{SWIR}$. The slopes of the OV group behave similarly, but the dependence on NDVI$_{SWIR}$ and %OV is stronger compared to the CV group. An increase in vegetation percentage is expected correspond to a darkening of visible surface reflectance, and thereby decreases the Red vs. SWIR slope. The urban slope also decreases as NDVI$_{SWIR}$ increases, but as % urban increases, it increases due to the greater contribution of high reflectance from human-made structure. The slope is high when SZA is low for groups of OV and urban, but in the CV group, a weak positive change is shown.“*

[Figure]

*Figure 13. The slope of regression between Red and SWIR AC-ref. The slopes are calculated for three land types, and shows changes depending on (a) SZA, (b) NDVISWIR, and (c) %land type. For each group of Closed Vegetation (CV, Red), Open Vegetation (OV, Green), and Urban (Gray), AC-ref are equally divided into 12 bins according to the x-parameter, respectively. Among the 193 AERONET sites used for the collocation, 40 sites corresponded to CV, 169 sites to OV, and 55 sites to urban.*

- I recommend to try Spearman's coefficient. Because the latter is more sensitive to rank and this may help to determine the SRP's dependency parameters (For discussion only, the author does not need to revise this point).

Thank you for the suggestion. In this manuscript, the 'R' represents Pearson's coefficient and has not been replaced with the Spearman's coefficient at this point. However, we will consider utilizing the Spearman's coefficient in future studies.

- I recommend to describe the new SRP estimation method (as Table 1) with equations in detail.

Thanks for the comment. Following description was added on lines 545-549.

*"Taking an OV-dominated pixel as an example, based on the MLR coefficients provided in Table 1, the slope and y-intercept can be predicted using the following equations:*

$$Slope_{RedSWIR}^{OV} = 0.0014 \times SZA - 0.4477 \times NDVI_{SWIR} + 0.0001 \times \%OV + 0.6729 \qquad (13)$$

$$Slope_{RedSWIR}^{OV} = -0.0003 \times SZA + 0.0411 \times NDVI_{SWIR} - 0.0001 \times \%OV - 0.0041 \quad (14)$$

*For a given NDVI$_{SWIR}$ and %OV, the MLR yields high Red-to-SWIR ratio when SZA is low."*

- Line 547: Why not using Land type data with higher spatial resolution? Because the original L1B data is ~1km. If you remove some original pixels when aggregating retrieval pixel for DT (~10km), the removed pixels should not be used to calculate land type percent.

Thank you for bringing this issue. The spatial resolution of %land type follows the Urban Percentage (UP) in the Baseline algorithm (Gupta et al., 2016). The ancillary map of UP was the optimal choice in LEO observations, considering that the ground pixels covered by the grid box change with each observation. However, GEO observations require more attention on how to calculate land types in that they continuously detect the same area, and follow-up research should be conducted on this. Regarding this issue, we've added discussion on lines 671-673.

*"The approach to calculating %land type also requires improvements in that there is a discrepancy between map of %land type and the ground pixels observed from GEO sensor. Concerns about spatial discontinuity in the AOD retrieval according to categorization of land cover type need to be addressed as well. "*

- Figure 14: Yes, the key blue band surface reflectance underestimation leads to the overestimation of the overall AOD in the baseline test. But the time dependence of AOD bias still partially remains as Figure 16.

That is true. The new SRP alleviates the AOD bias when SZA is low but shows less improvement at dusk/dawn. We should pay more attention to address uncertainties in AC-ref calculations at high SZA in future studies. Since we take only the SZA into account, the vis vs. SWIR relationship changes according to the VZA and the RAA can result in the AOD bias also.

- Although figures 15-17 are useful to describe the results of this new method. However, there is still a lack of a total validation scatter plot to represent the overall retrieval results, error distributions, and before and after improvements, as many other studies have done.

We agree with this comment and have added DT AOD validations (Fig. 19 and description on lines 618-625).

*"The scatter plots in Fig. 19 compares the DT AODs with AERONET AOD. By applying the new SRP, the %EE has been improved from 50.53% to 75.57%, and the regression slope became closer to 1. Figure 20 shows the same plots as seen in Figure 17 but for three specific AERONET site, GSFC, NEON Harvard, and PNNL, which are dominated by Urban (44.09%), mixed forests (47.57%), and Croplands (86.87%), respectively. The AERONET sites are observed from ABIE with fixed VZA of 45.42°, 49.38°, and 68.99°, respectively. The new SRP reduces the AOD bias at the three AEROENT site but has less impact on the mitigation of the time-dependence at PNNL. The AOD retrieved at PNNL may have greater uncertainty in that the site is located at the high VZA near the boundary of ABIE's observing disk, and the cropland surface contributed to the bright reflectance. "*

[Figure]

*Figure 19 Comparison of the AEORNET AOD and the DT AOD retrieved from ABIE from August to September 2019 across all collocated AERONET stations within the sensors' disk scan. The DT AOD are retrieved by applying (a) the baseline and (b) the new SRP, respectively. The color indicates number density of scatter plot.*

Reviewer 2

This is a review for "Parameterizing spectral surface reflectance relationships for the Dark Target aerosol algorithm applied to a geostationary imager" from Kim et al. This manuscript extends the well known Dark Target (DT) aerosol retrieval algorithm to the geostationary (GEO) imager ABI on GOES-16 by modifying the so-called Surface Reflectance Parameterization (SRP). This key processing step of the DT algorithm allows it to constrain the spectral variation of surface reflectance as a previous step to the estimation of aerosol properties over dark surfaces. The topic of the manuscript is of high interest for the atmospheric community and very appropriate to AMT because GEO imagers such as ABI are becoming increasingly compelling for aerosol remote sensing thanks to their high revisit time and their increasingly advanced sensing performances. The experiments described in the manuscript are relevant and most of results are convincing. However, and although it reads quite well most of the time, the manuscript would benefit of a careful revision, especially in some sections that are a bit too long (with redundant information) and some others that despite being key are a bit unclear currently (e.g. Sect. 5.2.1). Furthermore, I am a bit doubtful on the application of the proposed solution to other GEO imagers (see major comment below), as I understand that the future goal is to apply DT to the current ring of GEO satellites in conjunction with the LEO heritage spacecraft. Please find some comments and suggestions to improve the manuscript before its acceptance in AMT.

Major comment:

- I agree that land surface reflectance in some VIS and SWIR wavelengths present some correlation with each other for vegetation and dark-soiled surfaces as described in Kaufman and Remer (1994). However, this correlation is known to be imperfect and although the current MODIS DT algorithm includes modifications of the original technique to account for the residual angular-dependent biases coming BRDF anisotropy (Levy et al., 2007) these inaccuracies seem to be much more complex to model for GEO data due to their specific geometric features. I acknowledge the effort of the authors to expand their SRP method to GEO data by adding dependencies to other variables, as this will make possible to have a consistent GEO-LEO aerosol product from DT. However, I have the feeling that this manuscript is showing the algorithmic limitations of SRP, as its adaptation to GEO seems to become a bit too much complex which may in turn compromise the robustness of DT. For example, I wonder how this technique will work for other GEO sensors (seeing land surfaces with other geometries and corresponding to other cover types) giving its dependency on the GOES-16 characteristics (already discussed by the authors in the "Conclusion" section). I would like the authors to elaborate on this point and on the potential ways of using a more general, less tailor-made methodology in the future. One idea would be to consider a full characterization of the spectral surface BRDF, which should be possible from GEO and especially with joint GEO-LEO data. One way forward could be to think of a "surface BRDF parameterization" instead of the current "surface reflectance parameterization". The consideration of the anisotropy of the surface reflectance might prevent the authors to consider an empirical method with so many dependencies as they are doing in their study. In this regard, I find interesting the discussion of the authors in P14 (L430-438) for example and think that this point would be worth discussing further.

This is very interesting discussion. We appreciate that. Yes, these are limitations, but the point is we are trying to see if we can reach consistency global GEO/LEO.

We have put your comments into the discussion (lines 666-670) as follows.

*"In this study, an improvement of SRP was achieved by applying the %land type parameterization, which distinguishes between open and closed canopy. However, it still does not fully characterize the surface BRDF, potentially resulting in residual angular bias in the surface reflectance and AOD. To ensure consistent DT retrievals across both LEO and GEO platforms, it is advisable to conduct a full characterization of the surface BRDF using a combined GEO-LEO AC-ref dataset. This dataset should cover a range of VZA, SZA, and RAA for each specific location."*

Other comments (of varying importance):

-P2, L58: HIMAWARI is not written in capital letters and please do not forget Himawari-9, which is the operational satellite since some months now.

Thanks for letting us know what we are missing. The lines 71-72 has been revised.

-L60-62: it would be great if you could explain why we want to retrieve aerosols from the GEO orbit. Which is the advantage? Did some studies put in evidence that LEO-derived aerosol observation are not sufficient?

The revised introduction includes the study of Kim et al. (2020) [doi.org/10.1175/BAMS-D-18-0013.1], emphasizing the advantages of GEO monitoring for both aerosols and trace gases (lines 64-69). In addition, we have included the findings of Saide et al. (2014) [doi.org/10.1002/2014GL062089] showing enhanced data assimilation achieved by applying GOCI AOD.

-Please consider mentioning other past and recent aerosol retrieval algorithms applied to GEO data, not only for ABI but also AHI, GOCI, SEVIRI, etc. GEO aerosol remote sensing is not something new!

As noted by the reviewer, significant efforts have been made for GEO aerosol retrieval and have been successful in providing continuous monitoring of aerosol. Various approaches have been implemented, including the Minimum Reflectance Approach for multiple sensors (e.g., AVHRR: Knapp et al., 2002; GOCI/GEO-KOMPSAT 1: Choi et al., 2018; MI/GEO-KOMPSAT 1: Kim et al., 2018), as well as a joint retrieval of AOD and surface reflectance has been developed for the MSG/SEVIRI observations based on the optimal estimation [e.g., Ceamanos et al., 2023, Govaerts et al., 2010]. Additionally, the DT approach has been successfully applied to AHI observations (Lim et al., 2018; Gupta et al., 2019) and ABI (Laszlo et al., 2022; Zhang et al., 2020).

Since the primary focus of this manuscript is to discuss enhancing the DT SRP for GEO AOD retrieval, our emphasis was on introducing the DT approach. Consequently, we have revised the

introduction (lines 72-87) to show the DT-branch studies [Lim et al., 2018; Gupta et al., 2019; Laszlo et al., 2022].

-P3, L71-73: Do "NOAA's aerosol products created from ABI" use SRP too? Why does this sentence appear here otherwise?

It's true. The NOAA aerosol product is retrieved based on the DT approach (Laszlo et al., 2022 [https://www. star. nesdis. noaa. gov/jpss/documents/ATBD/ATBD_EPS_Aerosol_AOD_v3 4]; Zhang et al., 2019). The reference has been added in line 74.

-L76: Raleigh -> Rayleigh

Thanks for the correction. It has been revised in line 80.

-P5, L138: new line is missing here

The sentence has been revised.

-Eq. 4 is missing "if NDVI..." in the three cases

Eq. 4 has been revised.

-P6, L163: "a" radiative transfer (RT) code

Thanks for the correction.

-L179: ".-The algorithm" -> Replace "-" by " "

Thanks for the correction.

-L193: What happens with SRP performances for cover types that are not represented by AERONET sites?

DT land algorithm masks out the bright surface, ice/snow, and in land water because the BRDF characteristics significantly differ from the DT assumption. Therefore, we excluded Permanent Wetlands (index 11), Permanent Snow and Ice (index 15), Barren (index 16), and Water bodies (index 17) from the SRP analysis.

In the retrieval process, 'Open Vegetation' which represents the most diverse surface type (IGBP index 5~10, 12, 14) is set as default for all available pixels. Then other types of SRPS are applied if the pixel is mainly covered by closed vegetation or urban.

-P7, L198: "which better matches the AERONET coverage observed by each ABI during 2019" Why 2019? You did not talk about the period of your study yet if I am not wrong.

The sentence was miswriting, and it has been revised in lines 185-187 as follows.

*"Therefore, to compare with the ABI datasets described in the next section, we perform AC on the subset of the AERONET sites that are observed by the corresponding ABI. We also constrain this analysis to the MODIS data (Aqua, MYD04 products) between 2015-2019."*

-L206 and L210: ABI red band is at 0.64 or 0.63 um? And SWIR band, is it at 2.24 or 2.26 um?

Sorry for the confusion. The wavelength information has been revised throughout the manuscript.

-P8, L250: "AOD at 0.55 µm is less than 0.2" -> what happens with sites with generally higher AOD values? Do you have enough data to characterize SRP properly at these sites? Won't be these sites underrepresented in the final regression?

Kim et al. (2021) [https://doi.org/10.1016/j.atmosenv.2020.117994] conducted a study on the global distribution of 'background AOD [BAOD]'. The study defines BAOD as the average of the lowest 5th percentile of AOD measured at each AERONET site. The result indicates that the maximum BAOD was 0.18 in South Asia, while in the remaining regions, BAOD remained below 0.10 throughout the year. Which means, in the most of AERONET sites, AOD lower than 0.1 accounts for at least 5% of total observations. Although the data periods are different, it was assumed that there would be sufficient observations to meet the criteria at most AERONET site.

The Fig.1 shows the locations in AERONET used for AC-ref calculations, with the colors indicating the number of datasets used. The map illustrates sufficient data pools within the ABIE FOV at each AERONET site.

[Figure]

*Figure 1. AERONET map with observations available for AC-ref calculation (AOD at 550 nm < 0.1, AE > 1.0) between July 2019 and June 2020. The color represents number of valid observation which meet the criteria and applied to the AC_ref calculation.*

-L250: "Ångström exponent (AE) (0.44 – 0.675 μm) is greater than" -> please complete and explain why you filter AERONET data according to the particle size.

The sentence has been corrected in lines 251-254. The AOD and AE is limited to avoid bias in AC-ref which can be caused by aerosol model assumption. Cases where coarse mode particle (AE<1.0) dominate are masked out to avoid large discrepancy in spectral AOD change between realistic and aerosol model assumption.

-L252: "The 193 AERONET site" -> sites. Where does the 193 figure come from? Is it the total of available sites? Or some selected sites? Showing a map of AERONET sites would be of interest here and would definitely help to better analyze Fig. 17 for which some stations seem to be missing.

The sentence has been modified in lines 374-376 to show the proportion of the data set for which 1640 nm observations are available. We found that 84% of the observations contained a valid 1640 nm AOD. The map displaying AERONET sites used for the AC-ref calculation was shown in Fig. 1 in the revised manuscript.

-L255 and L256: you talk first of ±0.3° and then of ±0.2° -> this is unclear

Collocation for AC-ref calculation and verification use different criteria for spatial matching. We used a wide spatial range for the TOA-ref collocation.    .

We've added a description on lines 256-265 as follows:

*"Spatiotemporal criteria for the satellite-AERONET co-locations are as follows. For AC-ref calculation, AERONET observations within ±15 minutes of satellite overpass (MODIS or ABIE) are collocated with satellite-derived TOA reflectance within ±0.3° rectangular grid centered over an AERONET site. Here, a relatively large spatial range was established to capture TOA reflectance influenced by diverse land cover types and to mitigate potential cloud contamination. The AC-ref dataset consists of the spatial mean of TOA reflectance, spectral AOD from AERONET, land cover types present with the range, and observation geometries. Figure 1 displays number of data used for the AC-ref calculation at each AERONET site. For the purpose of AOD validation, a temporal criteria of ±15 minutes for AERONET AOD and a spatial criterion of ±0.2° rectangular grid for DT AOD are applied. The spatiotemporal collocation window follows standard DT validation practice used in the MEaSUREs program."*

-P9, L278: did you describe "FD" previously?

It was changed to use the "Full Disk" instead of FD.

-Fig. 2: you do not specify the period of time covered by the data. Based on this figure, could you comment which geometry, LEO or GEO, is more suited to derive aerosol properties in principle?

Thanks for this comment. We added the description for the data period in the revised manuscript, Section 3.1. We also added the following discussion in lines 305-315 regarding difficulties in GEO AOD retrieval from the view of observation geometry.

*"Based on a comparison of observation geometries, it appears that LEO observation using the DT technique is more favourable to retrieve AOD than GEO observations. First, high SZA observations introduce greater uncertainty in AOD retrieval. Increased path length at high SZA makes it difficult to separate the aerosol contribution from other atmospheric components. The high SZA also can lead to uncertainty in AC-ref calculations and make the SRP analysis more difficult. Second, GEO observation are more likely to observe the 'vegetative hot-spot'. This is when the solar direction coincides with the observation direction ($\Theta \geq 175°$), resulting in a large increase surface reflectance at each wavelength. (Li et al., 2021) shows that ABI observe hot-spot frequently. The brighter surface overwhelms the aerosol contribution to the TOA reflectance and increases the uncertainty in AOD retrieval. Figure 3(g) also shows that ~4% of ABIE observations are performed at high scattering angles (>168°), whereas ~1% of MODIS observation reach that extreme at GSFC. The broad range of geometries lead to a difficulty in GEO AOD retrieval (Ceamanos et al., 2023) because the sensitivity to aerosols varies significantly during the day (Luffarelli and Govaerts, 2019; Ceamanos et al., 2019)."*

-P10, 294: "Although both scattering angle distribution in Fig. 2(g) and Fig. 2(h) cover the same angular range" -> I am surprised with this comment. I have always had the feeling that GEO cover a broader range than LEO during the year. Are bins too wide to be able to see that perhaps?

Figure 3 has been modified to have narrow bin, but it is still true that both scattering angles cover similar angle range. However, it should be emphasized that the GEO has more chance to sense at high scattering angle (>165°) than the LEO. Every day, there is noon (solar time) observation of the longitude sub-satellite point. Accordingly, GEO have more chance to observe near hot spot of surface bidirectional reflectance. The brighter surface can increase the retrieval uncertainty.

-P10, L302: "We must remember that for GEO sensors, a particular ground site is always observed with the same viewing angle while the sun angles change throughout the day." -> redundant

Thanks. We have removed the redundant the sentence.

-L304 "... and therefore each VZA matches up to a specific land cover type according to location" -> This is not true. VZA varies with distance to sub-satellite point, making concentric lines of equal VZA values.

We tried to explain that GEO observes a location with fixed VZA, like the VZA in GSFC is fixed at 45.42°, so VZA can be matched up with a land cover type of the location. We have revised the sentence to make it clear (lines 330-331).

*"For GEO sensors, a particular ground site is always observed with a fixed VZA (Fig. 2(a) and Fig. 3(e)) while the sun angles change throughout the time."*

-Fig. 3: the interpretation of the diurnal biases from the authors is arguable. Authors should try to explain why bias is greater at noon and dawn/dusk. And why the bias peak is shifted to the early afternoon for GOES-17? I recommend the authors to link these results, as well as others presented in their manuscript, to the well-known diurnal variation of aerosol information content in the case of GEO satellites (please see the works of Luffarelli et al., 2019 and Ceamanos et al., 2023; both in AMT).

Thank you for this comment. The references are very helpful. We aimed to mitigate the diurnal bias by modifying the surface reflectance parameterization, but time-dependent biases due to other factors may still be present. The diurnal change of AOD Jacobian shown in Luffarelli et al. (2019) and other discussions in Ceamanos et al. (2023) clarified this, so references were added to the revised manuscript (lines 313-315).

-P11, L338: which is the impact on SRP of considering a continental aerosol model only? Why not trying to use the DT models instead? This seems like a critical point to me.

Thanks for this suggestion. We agreed. The continental model assumption will introduce an error in AC-ref. In order to test the impact of aerosol model assumption, we used the aerosol optical properties obtained from AERONET inversion at the GSFC AERONET site [Appendix B]. The results show that different aerosol model assumptions lead to minor changes in the spectral relationship.

We believe this study should be expanded to include aerosol characteristics at a more diverse location. Based on the reviewers' comments, we acknowledge the need for further research to improve AC-ref accuracy. This should address concerns related to spatial resolution, uncertainties in high SZA conditions, and assumptions regarding aerosol models. This discussion has been added to lines 674-675.

-P12: What about using ground-based reflectance data to validate the retrieved AC-reflectance over a few AERONET sites?

Thanks for this suggestion. We agree that we need to assess the accuracy of AC-ref and make efforts to improve it. The AC-ref calculation in current approach has uncertainties regarding aerosol model assumption or plane-parallel atmosphere assumption in the LUT calculation. In this regard, previous DT study have used MODIS land surface reflectance (MOD09A1/MYD09A1) to investigate the SRP for urban (Gupta et al., 2016). In this study, we focused on evaluating the spectral relationship of the AC-ref and shortly discussed the uncertainty in AC-ref calculations, but we plan to pursue verification and improvement of AC-ref through future research.

-Fig. 5: How do you explain the variable precision of your AC-method according to SZA? Is higher SZA affected by higher multiple scattering, thus giving a lower precision?

That is true. AC-ref calculation is not expected to be highly accurate when SZA is high as we discussed above.

-P12, text on Fig.6 saying "modification of SRPs varying with SZA mitigates the bias peak at noon" -> this is right but we also observe a degraded accuracy at dusk/dawn. Diurnal evolution of bias stays similar, which may give you a spurious diurnal aerosol cycle at the end!

Agreed. The new SRP alleviates the AOD bias when SZA is low but shows no significant improvement at dusk/dawn. We should pay more attention to address uncertainties in AC-ref calculations at high SZA in future studies. Since we take only the SZA into account, the vis vs. SWIR relationship changes according to the VZA and the RAA can result in the AOD bias also.

-P13, L375-378: already said before

The repeated sentence has removed from the revised manuscript.

-L386: "there is increased scattering" -> do you mean "dispersion" or "noise"? How do you justify this increase?

The sentence has been revised (lines 410-412) to make is clear.

*"Figure 8(b) applies the same AC-ref technique but for the ABIE observations. While the slopes and y-offsets are overall similar to those observed when regressing MODIS, the visible and SWIR relationships are more scattered, with significantly reduced correlation coefficient."*

-L386: "with significantly reduced correlation in the blue/red relationship" -> Fig. 7 only seems to assess blue/SWIR and red/SWIR, so how do you conclude on blue/red correlation?

The sentence has been revised as follows (lines 410-412).

*"While the slopes and y-offsets are overall similar to those observed when regressing MODIS, the visible and SWIR relationships are more scattered, with significantly reduced correlation coefficient. "*

-L390: "newer MODIS data" and "the Western Hemisphere" -> unclear

The phrase has been revised in lines 414-415.

-P14, Fig.9: Would plots against SZA be clearer than against xi?

Thanks for this comment. We have added the changes with SZA (shown below) to Fig. 10, and described them in lines 442-444.

*"While the Red vs. SWIR regression shows weak dependence on the scattering angle, the regression changes linearly with SZA in Fig. 10(c) and Fig. 10(f). As SZA increases, the slope increases, and the interception decreases. The linear dependence is also evident in the blue-red relationship, although the points are slightly deviating from the linearity when SZA is high (>70°)."*

[Figure]

*Figure 10 Regression between visible and SWIR AC-ref of (left) MODIS and (middle and right) ABIE. Slope (a, b, c) and y-intercept (d, e, f) of regression are plotted as a function of (left and middle) scattering angle and (right) SZA. The data were sorted according to scattering angle/SZA and put into 20 groups of equal size (736 for MODIS and 3899 for ABIE). Blue square and red circle indicate red-SWIR and blue-red ratio, respectively.*

-L438: Sect. 5.2 title has not the right format

The format has been corrected.

-P15, L450: "In fact, the definition of BRDF requires a fixed SZA, and that SZA varies throughout every day.". Did you mean "that VZA varies..."? And why is that? Can you provide a reference on this affirmation regarding BRDF definition?

The sentence was deleted as it is unnecessary.

-Sect. 5.2.1 is key but not easy to follow, especially P16. Please consider improving/clarifying it.

Agree. Section 5.2.1 has been revised to remove redundant descriptions. AC-ref clustering for SRP analysis and land type assignment to each pixel during the retrieval process were also clarified.

-General comment on the LC-based methodology: What will happen in front of spatial changes of LC? Is there a risk to have spurious AOD boundaries?

We assumed that by applying the %land type, we would be able to prevent spatial inconsistency in AOD in front of LC change. So far, no notable AOD boundaries have been found. However, long-term searches will be necessary to track them, and more attention should be paid to how land types are calculated regarding the spatial resolution.

Regarding this issue, we've added discussion as follows (lines 671-673).

*"The approach to calculating %land type also requires improvements in that there is a discrepancy between map of %land type and the ground pixels observed from GEO sensor. Concerns about spatial discontinuity in the AOD retrieval according to categorization of land cover type need to be addressed as well. "*

-Fig. 16 is convincing but I would like to see the same plots for a few sites with different information content on aerosols (cad. different surface reflectance and location/geometry). For example, what happens over the GSFC site with respect to Fig. 6b? A site in a more arid area (Midwest?) would be great too. Also, please detail the period of time that was considered to generate this plot.

The following figures showing the same plots for Fig. 17 are obtained from three AERONET site, GSFC, NEON Harvard, and PNNL, where are dominated by Urban (44.09%), mixed forests (47.57%), and Croplands (86.87%), respectively. The AERONET sites are observed from ABIE with fixed VZA of 45.42°, 49.38°, and 68.99°, respectively. The new SRP reduces the AOD bias at the three AEROENT site but has less impact on the mitigation of the time-dependence at PNNL. The AOD retrieved at PNNL may have greater uncertainty in that the site is located at the high VZA near the boundary of ABIE's observing disk, and the cropland surface contributed to the bright reflectance.

[Figure]

*Figure 20. Comparison of the bias in the DT AOD at GSFC, NEON_Harvard, and PNNL AERONET site. The bias is calculated from the DT baseline and new retrieval from August to September 2019. The range of bias for each time is expressed as a vertical bar connecting the first and third quartiles., and the median bias is shown by symbol. Color indicates the baseline (Red) and the new AOD (Black).*

-Generally on the results, it would be interesting to investigate the link between surface reflectance change and AOD change (between baseline DT and new DT), and all this according to

scattering angle. I would not be surprised to see that AOD improves more for angles associated to a lower information content despite experiencing similar reflectance changes.

The following figures shows the link between surface reflectance change (at red) and AOD change [new – baseline] according to the scattering angle. In most case, the surface reflectance is increased by applying the new SRP, and thereby the AOD is decreased. The changes in AOD and surface reflectance are clearly distinguishable depending on the scattering angle. The AOD change is significant where the scattering angle is higher than ~150° despite the surface reflectance change is small.

*This figure has not added into the revised manuscript because the decreasing AOD and its angle dependence were already seen in the previously shown figures.*

[Figure]

*Difference between the new and baseline DT AOD corresponding to the changes in surface reflectance at Red. The DT AODs are compared at three AEORNET site, GSFC, NEON_Harvard, and PNNL between August and September in 2019. The color indicates scattering angle.*

-Sect. 6.3: I would like to see some AERONET/DT density scatter plots, with some average statistics.

We agree with this comment and have added DT AOD validations (Fig. 19 and description on lines 618-625).

*"The scatter plots in Fig. 19 compares the DT AODs with AERONET AOD. By applying the new SRP, the %EE has been improved from 50.53% to 75.57%, and the regression slope became closer to 1. Figure 20 shows the same plots as seen in Figure 17 but for three specific AERONET site, GSFC, NEON Harvard, and PNNL, which are dominated by Urban (44.09%), mixed forests (47.57%), and Croplands (86.87%), respectively. The AERONET sites are observed from ABIE with fixed VZA of 45.42°, 49.38°, and 68.99°, respectively. The new SRP reduces the AOD bias at the three AEROENT site but has less impact on the mitigation of the time-dependence at PNNL. The AOD retrieved at PNNL may have greater uncertainty in that the site is located at the high VZA near the boundary of ABIE's observing disk, and the cropland surface contributed to the bright reflectance. "*

[Figure]

*Figure 19 Comparison of the AEORNET AOD and the DT AOD retrieved from ABIE from August to September 2019 across all collocated AERONET stations within the sensors' disk scan. The DT AOD are retrieved by applying (a) the baseline and (b) the new SRP, respectively. The color indicates number density of scatter plot.*

-Fig. 17: why some sites along the US East coast are not there while they are seen in the plots for GOES-17? Please comment on the missing sites and consider adding a map with all the considered sites to be able to see which are missing in each case.

Since bias calculations are performed on AERONET sites that have a valid number of retrieval (>20) for both the baseline and the new products during the testing period (August-September 2019), the operating site and the site shown on the map are differ. The US East coast is out boundary of ABIW DT retrieval because the VZA is greater than 72°. For the same reason, the number of AERONET sites in South America is also smaller.